# CortiLife: A Unified Framework for Cortical Representation Learning across the Lifespan

**Pengcheng Xue**[1,2]**, Dong Nie**[3]**, Jie Luo**[1,2]**, Daoqiang Zhang**[1,2] **& Xuyun Wen**[1,2] *

[1]College of Artificial Intelligence, Nanjing University of Aeronautics and Astronautics,
  Nanjing, Jiangsu, China
[2]Key Laboratory of Brain-Machine Intelligence Technology, Ministry of Education,
  Nanjing, Jiangsu, China
[3]Meta Inc., CA, USA
`charles1231@nuaa.edu.cn, dongnie@cs.unc.edu, sz2416070@nuaa.edu.cn`
`dqzhang@nuaa.edu.cn, wenxuyun@nuaa.edu.cn`

## Abstract

The human cerebral cortex encodes rich neurobiological information that is essential for understanding brain development, aging, and disease. Although various cortical representation learning methods have been proposed, existing models are typically restricted to stage-specific cohorts and lack generalization across the lifespan. While recent vision-language models offer a promising direction, building a unified framework for cortical representation faces three key challenges: (1) the non-Euclidean manifold structure of cortical surfaces, (2) homogenization of individual folding patterns induced by registration, and (3) distribution shifts of cortical features across the lifespan. To address these issues, we present CortiLife, the first unified vision-language framework for lifespan-aware cortical representation learning. Specifically, CortiLife introduces a surface tokenizer that integrates icosahedron-based surface patchification with multi-level patch encoding to transform complex cortical manifolds into compact token representations. The multi-level encoding incorporates three complementary streams that capture local topology, global interactions, and patch-wise distributional patterns, effectively mitigating the challenges of homogenization and distribution shifts. Furthermore, CortiLife integrates masked self-distillation with metadata language prompting, embedding information such as age, sex, health status, and attribution type into the text encoder to better capture individual-specific cortical representations while enabling both age-aware and modality-aware modeling. Extensive experiments on downstream tasks, including two encoder-frozen tasks (age prediction and cortical parcellation) and four encoder fine-tuning tasks (brain disorder diagnosis), demonstrate that CortiLife consistently outperforms state-of-the-art baselines across different age stages and modality types, underscoring its effectiveness and generalization ability.

## 1 Introduction

Representations of the cerebral cortex, which encode rich neurobiological information in metrics such as cortical thickness (CT), surface area (SA), and mean curvature (MC), are critical for both cognitive neuroscience and clinical diagnostics Hettwer et al. (2022); Storsve et al. (2014); de Vareilles et al. (2023). These structural features serve as powerful biomarkers, offering insights into neurodevelopmental and aging trajectories Dickerson & Wolk (2012); Fjell et al. (2015), as well as indicating pathologies associated with disorders like Autism Spectrum Disorder (ASD) Ecker et al. (2013) and Attention-Deficit/Hyperactivity Disorder (ADHD) You et al. (2024). Thus, learning effective cortical representations is essential for advancing personalized brain mapping and enabling computer-aided disease detection.

---

*Corresponding Author

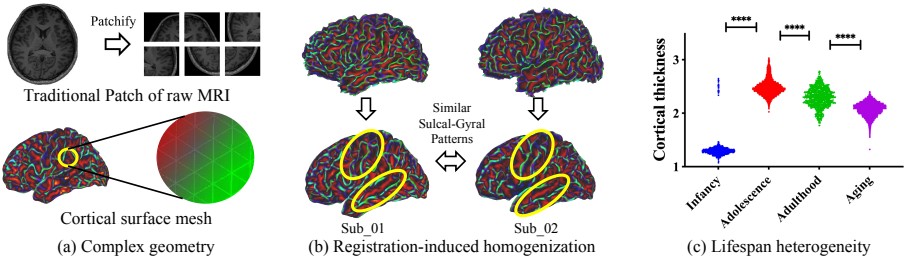

Figure 1: Challenges in universal cortical representation: (a) Non-Euclidean manifold structure; (b) Registration-induced homogenization; (c) Distribution shifts across the lifespan.

Recent advances in deep learning have introduced powerful tools for this task. Spherical CNNs Zhao et al. (2019; 2021) have been developed to respect the spherical topology of the cerebral cortex, while Surface Vision Transformers Dahan et al. (2022) have shown success in modeling long-range dependencies across cortical patches. Despite their progress, they share a critical limitation. These models are typically trained on narrow age cohorts, making them unable to account for the profound structural dynamics that the brain exhibits across the human lifespan. Therefore, developing a unified framework for learning cortical representations that bridge diverse developmental stages across the lifespan remains a key and unsolved challenge.

Concurrently, vision-language models (VLMs), exemplified by CLIP Radford et al. (2021), have demonstrated remarkable transfer learning capabilities by aligning large-scale image and text representations. This paradigm has also shown great promise in medical imaging, with models such as BiomedCLIP Zhang et al. (2023) and subsequent work Petersen et al. (2025); Lai et al. (2024); Cui et al. (2025) achieving state-of-the-art performance by linking visual data with textual medical records. However, extending this success to the cortical surface data is non-trivial, facing the following three challenges: (i) **Non-Euclidean manifold structure**: As shown in Figure 1(a), the cerebral cortex is a highly folded non-Euclidean manifold with intricate topological and geometric properties. This structure fundamentally differs from conventional 2D grids or 3D volumetric data, making standard convolution-based or grid-based vision models ineffective at directly capturing cortical geometry and local spatial patterns. (ii) **Registration-induced homogenization**: Standard preprocessing pipelines typically register individual cortical surfaces to a common template to enable cross-subject comparability, as shown in Figure 1(b). However, this registration inevitably reduces the distinctiveness of individual gyral-sulcal folding patterns, resulting in higher similarity of corresponding patches across subjects and thus hindering individualized representation learning. (iii) **Distribution shifts across the lifespan**: The cerebral cortex undergoes dynamic and complex structural changes across the human lifespan. These developmental variations give rise to diverse distributional shifts in different cortical features (e.g., CT, SA, and MC) at different age stages (as shown in Figure 1(c)). This poses significant challenges for achieving unified lifespan-aware cortical representation learning.

To address these challenges, we propose **CortiLife**, the first unified vision-language framework for cortical representation learning across the entire lifespan. The core components of CortiLife are a surface tokenizer and a vision-language model (VLM). The surface tokenizer integrates icosahedron-based surface patchification with multi-level patch encoding to transform complex cortical manifolds into compact token representations. In particular, the multi-level patch encoding module incorporates three complementary streams that jointly capture local topology, global interactions, and patch-wise distributional patterns, thereby mitigating the difficulties of representation learning caused by registration-induced homogenization and lifespan-related distribution shifts. For vision-language modeling, we adopt masked self-distillation representation learning as the visual backbone and introduce metadata language prompting, embedding information such as age, sex, health status, and feature type into the text input. This design allows the model to better capture individual-specific cortical representations while enabling both development-aware and feature-aware modeling. Extensive experiments on three primary surface-based tasks, including brain disorder classification, age prediction, and cortical parcellation, consistently show the state-of-the-art

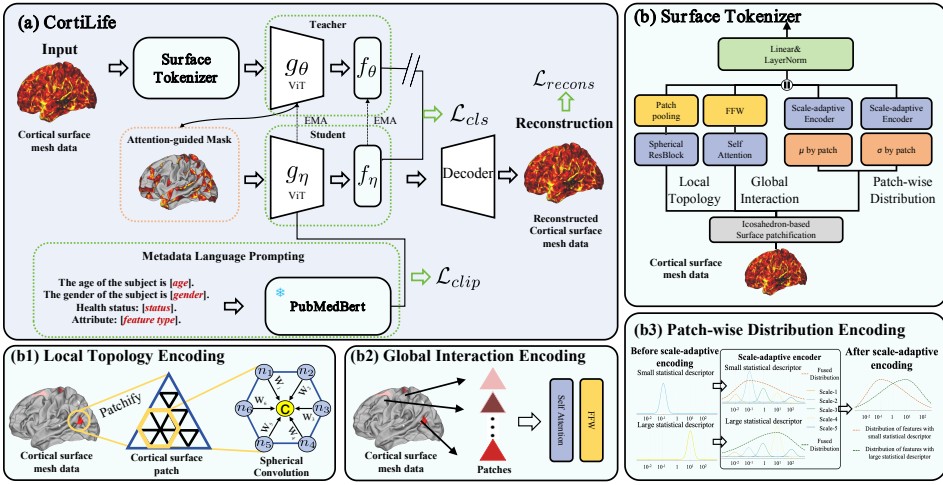

Figure 2: Overview of the CortiLife framework, which is composed of three main components: surface tokenizer, masked self-distillation learning, and metadata language prompting.

performance, demonstrating superior generalization across the cortical modalities and the entire lifespan.

## 2 RELATED WORK

### 2.1 CORTICAL REPRESENTATION LEARNING METHOD

The cortical surface is a highly folded non-Euclidean manifold, which conventional 2D and 3D CNNs cannot effectively model. To address this, Spherical U-Net Zhao et al. (2019) introduced convolution and pooling operations on resampled cortical meshes, achieving promising results in parcellation and developmental mapping. Its extension, Spherical Deformable U-Net Zhao et al. (2021), further incorporated deformable convolutions to better capture folding variability. More recently, transformer-based models Dahan et al. (2022); Cho et al. (2022) have leveraged self-attention on the sphere, advancing tasks such as infant cortical surface quality assessment and outperforming CNN-based baselines. Nevertheless, these approaches remain limited to specific cohorts or developmental stages, lacking the ability to generalize across the entire lifespan.

### 2.2 VISION-LANGUAGE MODEL

Contrastive Language-Image Pre-training (CLIP) Radford et al. (2021) has established a foundation for multimodal learning by enabling robust image-text alignment and generalization. In medical imaging, CLIP-style pretraining has been adapted to modalities such as X-ray Wang et al. (2022b); You et al. (2023), MRI Avci et al. (2025), and CT You et al. (2025), showing promise in tasks like report generation Liu et al. (2024) and zero-shot inference Huang et al. (2021). Large-scale efforts such as UniMed-CLIP Khattak et al. (2024) further extend CLIP to diverse modalities, while segmentation-oriented extensions like CLIPSeg Lüddecke & Ecker (2022) and CRIS Wang et al. (2022a) have broadened its applications. Nevertheless, these approaches remain focused on 2D or volumetric images, leaving the non-Euclidean cortical surface modeling problem largely unexplored.

## 3 METHODOLOGY

In this section, we describe the proposed CortiLife, with the overall framework illustrated in Figure 2. It is composed of three core modules: 1) a surface tokenizer, 2) masked self-distillation vision representation learning, and 3) metadata language prompting.

## 3.1 Surface Tokenizer

Before constructing the VLM, it is crucial to transform cortical surface data into compact token representations. Unlike conventional patchification methods for 2D or 3D images in Euclidean space, cortical surfaces lie on a non-Euclidean manifold, making them incompatible with regular square or cubic partitioning. Furthermore, cortical representation learning faces two major challenges: registration-induced homogenization and distributional shifts across different ages and modalities, which prevent tokenization strategies designed for natural images from producing effective representations. To overcome these issues, we introduce a surface tokenizer composed of two key modules: an icosahedron-based surface patchification module and a multi-level patch encoding module. In our work, we denote the cortical surface data as $x_c \in \mathbb{R}^{N_v * 1}$, where $N_v$ is the number of vertices. The value at each vertex encodes a morphological attribute, such as CT, MC, or SD. A cortical map containing a specific attribute is referred to as a cortical modality.

### 3.1.1 Icosahedron-based Surface patchification

Unlike Euclidean 2D images or 3D volumetric data, the cortical surface map cannot be partitioned into regular square or cubic patches. To address this, we adopt an icosahedron-based subdivision strategy, which divides the surface into local triangular facets and subsequently aggregates them into regular triangular patches Cho et al. (2022). Specifically, given cortical data with $N_v$ vertices, we reorganize all surface vertices such that each patch corresponds to a triangular structure containing an equal number of vertices, thereby forming $P$ patches. Through this process, the original cortical map $x_c \in \mathbb{R}^{N_v * 1}$ is partitioned into a triangular patch set $x_p \in \mathbb{R}^{P * N_p}$, where $N_p$ represents the number of vertices in each patch. In this work, we set $P = 640$, meaning that each hemisphere is partitioned into 320 patches.

### 3.1.2 Multi-level Patch Encoding

After patchification, we design a multi-level patch encoding strategy with three complementary streams that capture local topology, global interactions, and patch-wise distributional patterns for each patch, thereby mitigating the challenges of registration-induced homogenization and distributional shifts across the lifespan during the representation learning.

**Local Topology Encoding.** The local topology encoding aims to focus on capturing fine-grained geometric and morphological cues within each patch, thereby preserving localized cortical details. Such fine-grained representations establish a foundation for identifying subtle variations across individuals and developmental stages. Specifically, we employ a spherical convolution-based surface encoder, termed Spherical ResBlock, to capture localized spatial information within each patch Zhao et al. (2019). This module is composed of an initial stem layer, four batch normalization layers, and four spherical convolution layers with residual connections. For each input triangular patch $x_p \in \mathbb{R}^{1 * N_p}$, it produces the corresponding embedding $e_p^L \in \mathbb{R}^{1 * M_v}$, where $M_v$ represents the feature dimensions of local topology encoding.

**Global Interaction Encoding.** Global interaction encoding aims to model long-range dependencies across patches, enabling the representation to integrate broader contextual information across the cortical surface. Specifically, we employ four layers of self-attention followed by four feedforward blocks to capture global interaction representations across all the patches. Finally, this module produces the corresponding global interaction representations for each patch, denoted as $e_p^G \in \mathbb{R}^{1 * M_p}$, where $M_p$ represents the feature dimensions of global interaction encoding. These representations provide a holistic perspective for characterizing structural variations of the brain across the entire lifespan and across individuals.

**Patch-wise Distribution Encoding.** Considering the feature distribution shifts across different age groups and different modalities as illustrated in Figure 1(c), we specially design a patch-wise distribution encoding strategy, named scale-adaptive encoder. As shown in Figure 2(b3), its core idea is to introduce a set of scale bases, each corresponding to a different statistical level of the feature distribution. Patch representations from all ages are projected onto these shared scale bases and then fused adaptively based on their original statistics. This design simultaneously unifies cross-age distribution levels and preserves age-specific distribution characteristics. The detailed implementation process is described as follows. First, it computes statistical descriptors $x_m$, including the mean ($x_{pm} \in \mathbb{R}^{P * 1}$) and standard deviation ($x_{ps} \in \mathbb{R}^{P * 1}$), across all the vertices within each patch, and

adaptively projects them into $n$ scale spaces by using the following formulation:

$$\mathbf{z}_i(x_m) = LN(x_m \cdot \mathbf{w}_i + k_i \cdot \mathbf{b}_i), i \in [1, ..., n], x_m \in [x_{pm}, x_{ps}] \tag{1}$$

where $LN$ denotes Layer Normalization, $k_i$ is a predefined scale value, $\mathbf{w}_i$ and $\mathbf{b}_i$ are learnable weighting and bias parameters at the $i$-th scale space. The term $k_i * b_i$ serves as a scale-dependent bias that maps the mean and variance descriptors into distinct scale spaces. Through Equation 1, we obtain the representation of distributional descriptors $\mathbf{z}_i$ of $x_m$ at the $i$-th scale space. Second, we design a scale-adaptive weighting mechanism to aggregate these representations across $n$ scales, which is formatted as

$$\mathbf{y}(x_m) = \sum_{i=1}^{n} \alpha_i(x_m) \cdot \mathbf{z}_i(x_m), \quad \text{where} \quad \alpha_i(x_m) = \frac{\left| \log^{-1}\left( \frac{|x_m|}{k_i} + \epsilon \right) \right|}{\sum_{j=1}^{n} \left| \log^{-1}\left( \frac{|x_m|}{k_j} + \epsilon \right) \right|}. \tag{2}$$

where the weighting function $\alpha_i(.)$ implements a gating mechanism over a set of predefined scales $\{k_1, ..., k_n\}$. The weighting function $\alpha_i(.)$ integrates the representations $\mathbf{z}_i(x_m)$ according to $n$ scales to learn embeddings in multi-scale distribution space, thereby mitigating diverse age-related distribution shifts. In this step, scales that are more compatible with the patch's intrinsic distribution are assigned larger weights, enabling the encoder to better preserve the individual statistical patterns. By using the above two steps, we obtain the final patch-level distribution pattern representation $e_p^S \in \mathbb{R}^{\tilde{P} * M_s}$, where $M_s$ represents the feature dimensions of distribution encoding.

**Embedding Fusion.** After obtaining patch representations at three different levels, we concatenate them along the channel dimension and pass the concatenated features through a linear layer to enable channel-wise interactions, yielding the final output of the surface tokenizer for each patch, denoted as $e_{tokenizer} \in \mathbb{R}^{1*(M_v+M_p+M_s*2)}$. In this work, the $M_v$, $M_p$, and $M_s$ are set to 256, 256, and 32, respectively.

## 3.2 MASKED SELF-DISTILLATION VISION REPRESENTATION LEARNING

Studies have shown that the cerebral cortex exhibits substantial spatial redundancy Zhao et al. (2023), which motivates our use of an MAE-based strategy for representation learning. However, our goal is not only cortical reconstruction but also semantic alignment with metadata through a CLIP-based objective. While random masking is effective for reconstruction, it does not ensure that developmentally informative regions remain visible, which is critical for effective alignment. To overcome this limitation, we adopt a masked self-distillation strategy in a teacher-student framework, where the teacher provides semantically enriched attention guidance and the student focuses on patches carrying high-level developmental information. This design allows the model to achieve both high-quality reconstruction and more effective developmental semantic alignment. Specifically, both teacher and student networks share the same 10-layer Transformer architecture, with the teacher updated through the exponential moving average (EMA) of the student. The teacher network processes the full set of patch tokens to capture holistic cortical representations, whereas the student network learns from a masked subset of cortical patches. The masked patches in the student network are determined based on self-attention scores computed from the representations generated by the teacher. Concretely, given the learned representations from the teacher network, we compute an attention score for each patch as follows:

$$AttScore_j = \frac{1}{H} \sum_{i=1}^{H} Softmax\left( \frac{Q_i \cdot K_i(j)}{\sqrt{d}} \right), \tag{3}$$

where $H$ denotes the total number of attention heads across 10 ViT blocks, $Q_i$ is the query vector of the [CLS] token of the $i$-th attention head, $K_i(j)$ represents the key vector of patch $j$ of the $i$-th head. We use this development-aware [CLS] (optimized by following metadata language prompting) as a query over patch tokens and retain regions with the highest 25% attention weights, ensuring that the student networks learn from the most informative cortical regions. The global embeddings obtained from the teacher and student networks are denoted as $E_{teacher}$ and $E_{student}$. For the detailed discussion of the effectiveness of this component, please refer to Appendix A.4.

In this module, we design two loss functions, including reconstruction loss $L_{recons}$ and alignment loss $L_{cls}$. The reconstruction loss $L_{recons}$ is employed to ensure that the student network accurately

recovers the fine-grained details of cortical data, and is defined as:

$$\mathcal{L}_{recons} = \frac{1}{|\mathcal{M}|} \sum_{i \in \mathcal{M}} |\hat{x}_i - x_i|^2 + \frac{1}{|\mathcal{V}|} \sum_{i \in \mathcal{V}} |\hat{x}_i - x_i|^2, \tag{4}$$

where $\mathcal{M}$ and $\mathcal{V}$ denote vertices of masked and visible patches.

Alignment loss, $L_{cls}$, is designed to maintain global semantic consistency between $E_{teacher}$ and $E_{student}$, since both networks are trained to extract features from the same cortical data. In this work, we employ the KL divergence to enforce distributional alignment between the two representations, which is defined as:

$$\mathcal{L}_{cls} = \lambda \cdot \mathrm{KL}\Big(p^{(t)} \parallel p^{(s)}\Big), \tag{5}$$

where $p^{(t)}$ and $p^{(s)}$ represent the distribution of $E_{teacher}$ and $E_{student}$, respectively.

### 3.3 METADATA LANGUAGE PROMPTING

To ensure robust generalization across the lifespan and different modalities, we incorporate lifespan-aware metadata (age, sex, health status, and feature type) into the training process, guiding the vision encoder to capture high-level developmental semantics. For semantic modeling, we adopt PubMed-BERT Gu et al. (2021), a domain-specific language encoder pretrained on large-scale biomedical text. By providing a fixed semantic space, PubMedBERT generates discriminative embeddings for similar metadata and enhances the alignment of developmental information. The textual input is formulated using the template: "*The age of the subject is [age]. The gender of the subject is [gender]. Health status: [status]. Attribute: [feature type].*".

For the objective of vision-language modeling, we employed the classical contrastive learning loss function. The loss function is defined based on the cosine similarity between a vision embedding $E^i_{student}$ and a text embedding $T_j$, formulated as:

$$logits_{i,j} = \frac{s_{i,j}}{\tau}, \ where \ s_{i,j} = \frac{E^i_{student} \cdot T_j}{\|E^i_{student}\| \, \|T_j\|} \tag{6}$$

The logits will be scaled by a learnable temperature $\tau$. And the loss function of image-to-text matching (i.e., $\mathcal{L}_{I2T}$) and text-to-image matching (i.e., $\mathcal{L}_{T2I}$) are defined as follows.

$$\mathcal{L}_{I2T} = -\frac{1}{N} \sum_{i=1}^{N} \log\left(\frac{\exp(logits_{i,i})}{\sum_{j=1}^{N} \exp(logits_{i,j})}\right), \mathcal{L}_{T2I} = -\frac{1}{N} \sum_{i=1}^{N} \log\left(\frac{\exp(logits_{i,i})}{\sum_{j=1}^{N} \exp(logits_{j,i})}\right) \tag{7}$$

Finally, we compute the average loss of $\mathcal{L}_{I2T}$ and $\mathcal{L}_{T2I}$.

$$\mathcal{L}_{clip} = (\mathcal{L}_{I2T} + \mathcal{L}_{T2I})/2 \tag{8}$$

### 3.4 TOTAL LOSS FUNCTION

The total loss combines the reconstruction loss $\mathcal{L}_{recons}$ and the alignment loss $\mathcal{L}_{cls}$ from masked self-distillation vision representation learning, and the image-to-text matching loss $\mathcal{L}_{clip}$, which are formatted as

$$\mathcal{L} = \mathcal{L}_{clip} + \mathcal{L}_{cls} + \mathcal{L}_{recons} \tag{9}$$

## 4 EXPERIMENTS AND RESULTS

### 4.1 DATASETS

In this study, we collected large-scale imaging data spanning the entire lifespan for model training and evaluation, comprising nine datasets in total. The overall statistics of each dataset are summarized in Table 1. In this study, we used three representative cortical modality data as examples for model generalization evaluation, including cortical thickness (CT), mean curvature (MC), and sulcal depth (SD). After preprocessing, each cortical data set contains 81,924 vertices in total, with 40,962 vertices in each hemisphere. More detailed dataset descriptions are provided in the Appendix A.2.

Table 1: Summary of dataset information.

| ID | Dataset | Samples | Age | Gender (M/F) | Diagnosis (case/control) |
|----|---------|---------|-----|--------------|--------------------------|
| 1 | DHCPMakropoulos et al. (2018) | 887 | 26-45 weeks | 478/409 | 0/887 |
| 2 | CHD | 229 | 1-3 years | 123/106 | 128/101 |
| 3 | CBCPXu et al. (2024) | 252 | 1-6 years | 133/119 | 0/252 |
| 4 | CCNPFan et al. (2023) | 559 | 4-18 years | 304/255 | 0/559 |
| 5 | ADHD-200Bellec et al. (2017) | 972 | 7-27 years | 599/373 | 388/584 |
| 6 | ABIDE II | 1,114 | 5-64 years | 856/258 | 521/593 |
| 7 | ABIDE I | 1,112 | 6-64 years | 948/164 | 539/573 |
| 8 | HCPVan Essen et al. (2013) | 1,206 | 22-36 years | 550/656 | 0/1,206 |
| 9 | ADNIJack Jr et al. (2008) | 7,597 | 55-95 years | 4,185/3,412 | 5,438/2,159 |
| 10 | Total | 13,928 | 26(w)-95(y) | 8176/5752 | 7014/6914 |

Table 2: Performance comparison on downstream tasks under the encoder-frozen setting.

| Task | (a) Age prediction | | | (b) Cortical parcellation | | |
|------|------|------|------|------|------|------|
| | MAE | | | DICE | | |
| Methods | CT | MC | SD | CT | MC | SD |
| CLIP | 3.603±0.319 | 3.228±0.341 | 3.204±0.119 | 0.721±0.028 | 0.760±0.018 | 0.919±0.007 |
| ACLIP | 3.243±0.020 | 3.193±0.106 | 3.155±0.052 | 0.636±0.006 | 0.672±0.015 | 0.745±0.027 |
| DetailCLIP | 3.156±0.105 | 3.112±0.047 | 3.137±0.091 | 0.785±0.008 | 0.804±0.006 | 0.832±0.014 |
| CARZero | 5.682±1.463 | 5.195±1.816 | 5.914±1.592 | - | - | - |
| CortiLife | **3.124±0.078** | **2.990±0.120** | **3.006±0.119** | **0.905±0.005** | **0.925±0.003** | **0.957±0.001** |

## 4.2 EXPERIMENTAL SETUP

**Environmental setup.** For pretraining, CortiLife and all other pretrained baseline models were pretrained using AdamW (lr=5e-4, weight decay=1e-4) with batch size of 64 for 10 epochs on four NVIDIA 3090 GPUs. For the downstream fine-tuning, we employed the stochastic gradient descent (SGD) optimizer with a learning rate of 0.001 and a batch size of 40, and conducted on a single NVIDIA 3090 GPU for 200 epochs. The selection of mask ratio in masked self-distillation framework is shown in the Table 8 in Appendix.

**Encoder Frozen on Downstream Tasks.** All methods were pretrained on eight datasets, including DHCP, CHD, CBCP, CCNP, ADHD-200, ABIDE-II, ABIDE-I, and ADNI. HCP was used for the two downstream-task evaluations, including age prediction and cortical parcellation. We kept the vision encoder fully frozen and trained only the MLP head for age prediction and cortical parcellation. For the training of MLP module, we use 80% of the data for training and the remaining 20% for testing. Performance for age prediction was evaluated using mean absolute error (MAE), while cortical parcellation was benchmarked against the DKT-40 ground truth using the DICE coefficient.

**Encoder Fine-tuning on Downstream Tasks.** We evaluate the model by encoder fine-tuning on four brain disease diagnosis tasks spanning different age groups, including CHD dataset(Congenital heart disease(CHD) vs Healthy controls), ABIDE I (Autism Spectrum Disorder (ASD) vs Healthy controls), ADHD-200(Attention Deficit/Hyperactive Disorder(ADHD) vs Healthy controls) and ADNI(Alzheimer's disease(AD) vs Healthy controls). In this experiment, the baselines include both non-pretrained and pretrained models. For the non-pretrained models, we use 80% of the data for training and the remaining 20% for testing. For the pretrained models, we first pretrain on all datasets except the target dataset, and then fine-tune on 80% of the target dataset while reserving the remaining 20% for testing. All models with pretraining share the same setting about pretraining datasets and downstream dataset. Accuracy (ACC) and area under the ROC curve (AUC) are used as evaluation metrics. In addition, we evaluate the performance of CortiLife with different proportions of data for fine-tuning, including 20% and 40% with results reported in Table 9 in the Appendix.

## 4.3 ENCODER FROZEN ON DOWNSTREAM TASKS

In this section, we evaluate the performance of CortiLife on downstream tasks under frozen-vision-encoder settings. We benchmark our framework against four state-of-the-art baselines, including CLIP Radford et al. (2021), ACLIP Yang et al. (2023), DetailCLIP Monsefi et al. (2024), and CARZero Lai et al. (2024). Due to issues in CARZero's alignment mechanism, we did not include it as a baseline for the parcellation task. All methods were implemented under the same experimental

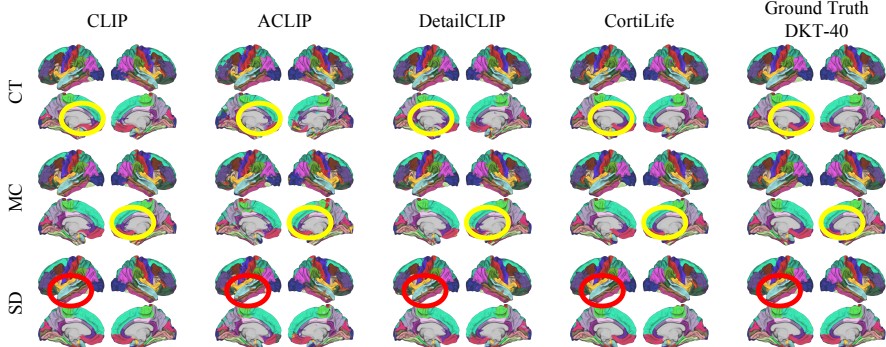

Figure 3: Visualization comparison of cortical parcellation. Our method produces results that are clearly superior to other approaches in both regional accuracy (e.g., the cingulate gyrus highlighted in yellow circles) and boundary details (e.g., the inferior temporal gyrus highlighted in red circles).

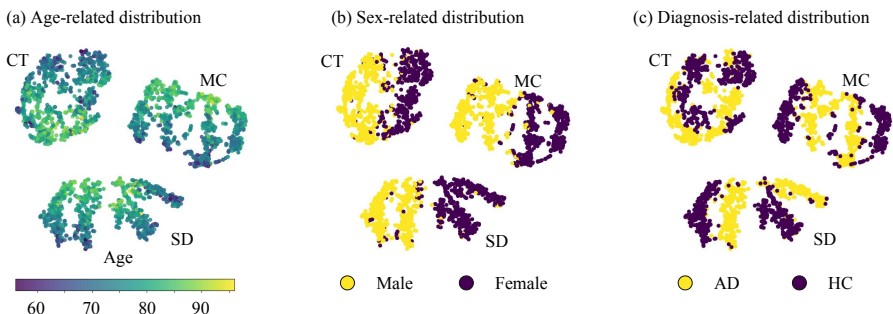

Figure 4: Visualization of representation embeddings from CortiLife in the ADNI dataset.

setup as ours for a fair comparison. The results are summarized in Table 2. In addition, we further visualize the embedding distributions to assess their effectiveness.

**Age Prediction.** As shown in Table 2(a), CortiLife achieves MAE values of 3.124, 2.990, and 3.006 on CT, MC, and SD, respectively, surpassing all baselines. These results indicate that the embeddings produced by CortiLife effectively capture structural characteristics associated with age information, thereby providing a faithful representation of the dynamic processes underlying brain changes.

**Cortical Parcellation.** Table 2(b) shows that CortiLife achieves DICE scores of 0.905, 0.925, and 0.957 on CT, MC, and SD modalities, respectively, surpassing the second-best models by 0.120, 0.121, and 0.038. In addition, the visualization results in Figure 3 demonstrate that cortical parcellations derived from our model's representations exhibit substantially higher accuracy in both regional delineation and boundary localization compared to SOTA methods. This result indicates that the embeddings learned by our model capture more fine-grained information and exhibit higher fidelity.

**Visualization of Embeddings.** For the pretrained model, we obtain the embeddings by feeding only the cortical surface features into the vision encoder, without providing any text prompts. We employ t-SNE to visualize the distribution of representations learned by CortiLife. Using the ADNI dataset as an example, we illustrate the representational differences across different age groups, genders, and disease categories. The results are shown in Figure 4. Figure 4(a) demonstrates that our method successfully captures age-related information for each cortical modality data, exhibiting a smooth gradient along the age continuum. Figures 4(b) and (c) further indicate that the model not only learns gender-specific heterogeneity in developmental patterns but also extends this heterogeneity to disease states, thereby highlighting sex-specific disease trajectories.

Table 3: Performance comparison on downstream brain disease diagnosis tasks under fine-tuning setting. CT, MC, and SD represent different cortical modality data.

| Modality | Methods | Pretrain | CHD (Infancy) ACC | CHD (Infancy) AUC | ADHD (Adolescence&Adult) ACC | ADHD (Adolescence&Adult) AUC | AD (aging) ACC | AD (aging) AUC |
|---|---|---|---|---|---|---|---|---|
| (a) CT | SphericalCNN | ✗ | 0.786±0.028 | 0.780±0.030 | 0.674±0.007 | 0.688±0.021 | 0.926±0.012 | 0.956±0.009 |
| | Spherical U-Net | ✗ | 0.787±0.021 | 0.789±0.029 | 0.668±0.018 | 0.668±0.018 | 0.917±0.009 | 0.965±0.007 |
| | WSSADN | ✗ | 0.792±0.021 | 0.821±0.045 | 0.681±0.010 | 0.669±0.027 | 0.926±0.008 | 0.961±0.006 |
| | NeuroExplainer | ✗ | 0.760±0.047 | 0.812±0.030 | 0.666±0.027 | 0.679±0.021 | 0.832±0.012 | 0.885±0.024 |
| | SurfaceVisionTransformer | ✗ | 0.785±0.04 | 0.796±0.017 | 0.671±0.010 | 0.673±0.020 | 0.819±0.017 | 0.877±0.023 |
| | CLIP | ✔ | 0.790±0.052 | 0.814±0.010 | 0.681±0.010 | 0.694±0.016 | 0.92±0.005 | 0.978±0.012 |
| | ACLIP | ✔ | 0.770±0.012 | 0.795±0.019 | 0.652±0.015 | 0.648±0.033 | 0.922±0.011 | 0.975±0.007 |
| | DetailCLIP | ✔ | 0.696±0.089 | 0.821±0.020 | 0.637±0.024 | 0.641±0.029 | 0.910±0.010 | 0.980±0.005 |
| | CARZero | ✔ | 0.777±0.001 | 0.797±0.006 | 0.628±0.006 | 0.604±0.027 | 0.908±0.025 | 0.979±0.001 |
| | CortiLife | ✔ | **0.806±0.011** | **0.823±0.026** | **0.697±0.015** | **0.730±0.007** | **0.928±0.002** | **0.981±0.001** |
| (b) MC | SphericalCNN | ✗ | 0.622±0.012 | 0.665±0.057 | 0.616±0.009 | 0.607±0.039 | 0.928±0.013 | 0.957±0.014 |
| | Spherical U-Net | ✗ | 0.641±0.037 | 0.693±0.060 | 0.628±0.018 | 0.616±0.020 | 0.927±0.005 | 0.960±0.007 |
| | WSSADN | ✗ | 0.642±0.060 | 0.721±0.064 | 0.630±0.029 | 0.617±0.052 | 0.921±0.010 | 0.955±0.014 |
| | NeuroExplainer | ✗ | 0.569±0.027 | 0.605±0.076 | 0.628±0.010 | **0.647±0.016** | 0.755±0.069 | 0.805±0.062 |
| | SurfaceVisionTransformer | ✗ | 0.647±0.042 | 0.722±0.063 | 0.623±0.003 | 0.582±0.015 | 0.831±0.025 | 0.889±0.020 |
| | CLIP | ✔ | 0.576±0.037 | 0.555±0.072 | 0.595±0.021 | 0.586±0.023 | 0.918±0.010 | 0.973±0.008 |
| | ACLIP | ✔ | 0.578±0.044 | 0.609±0.065 | 0.585±0.058 | 0.601±0.030 | 0.920±0.005 | 0.968±0.004 |
| | DetailCLIP | ✔ | 0.614±0.046 | 0.665±0.078 | 0.611±0.010 | 0.606±0.003 | 0.916±0.015 | **0.987±0.002** |
| | CARZero | ✔ | 0.578±0.022 | 0.660±0.055 | 0.609±0.005 | 0.536±0.037 | 0.928±0.005 | 0.985±0.001 |
| | CortiLife | ✔ | **0.667±0.031** | **0.776±0.014** | **0.632±0.005** | 0.618±0.018 | **0.939±0.011** | 0.973±0.002 |
| (c) SD | SphericalCNN | ✗ | 0.672±0.045 | 0.719±0.033 | 0.626±0.021 | 0.592±0.024 | 0.925±0.010 | 0.958±0.010 |
| | Spherical U-Net | ✗ | 0.682±0.049 | 0.732±0.034 | 0.629±0.012 | 0.580±0.037 | 0.933±0.010 | 0.969±0.006 |
| | WSSADN | ✗ | 0.721±0.027 | 0.739±0.039 | 0.621±0.012 | 0.595±0.023 | 0.935±0.012 | 0.975±0.015 |
| | NeuroExplainer | ✗ | 0.629±0.058 | 0.792±0.016 | 0.644±0.013 | 0.627±0.023 | 0.753±0.011 | 0.827±0.004 |
| | SurfaceVisionTransformer | ✗ | 0.673±0.039 | 0.653±0.061 | 0.582±0.015 | 0.545±0.026 | 0.831±0.046 | 0.891±0.042 |
| | CLIP | ✔ | 0.614±0.063 | 0.694±0.026 | 0.643±0.023 | 0.653±0.007 | 0.956±0.005 | 0.985±0.007 |
| | ACLIP | ✔ | 0.629±0.025 | 0.614±0.016 | 0.592±0.062 | 0.621±0.031 | 0.942±0.016 | 0.986±0.007 |
| | DetailCLIP | ✔ | 0.696±0.055 | 0.732±0.044 | 0.614±0.022 | 0.616±0.036 | 0.948±0.011 | 0.989±0.001 |
| | CARZero | ✔ | 0.656±0.047 | 0.678±0.022 | 0.607±0.005 | 0.526±0.063 | 0.942±0.017 | 0.987±0.001 |
| | CortiLife | ✔ | **0.739±0.033** | **0.799±0.073** | **0.651±0.010** | **0.657±0.017** | **0.972±0.002** | **0.991±0.002** |

## 4.4 ENCODER FINE-TUNING ON DOWNSTREAM TASKS

We further evaluate the fine-tuning performance of CortiLife using disease classification as a case study. Experiments are conducted across different age groups (infants, adolescents, adult and elderly subjects) and morphological modalities (cortical thickness, mean curvature, and sulcal depth). We compare our framework against nine state-of-the-art baselines. Among them, five are non-pretrained methods, including Spherical CNN Zhao et al. (2019), Spherical U-Net Zhao et al. (2019), WS-SADN Xue et al. (2024), Surface Vision Transformer Dahan et al. (2022), and NeuroExplainer Xue et al. (2023); while four are pretrained + fine-tuning methods, including CLIP Radford et al. (2021), ACLIP Yang et al. (2023), DetailCLIP Monsefi et al. (2024), and CARZero Lai et al. (2024). All methods were implemented under the same experimental setup as ours for a fair comparison. Results for CHD, ADHD, and AD diagnosis are given in Table 3, and results for ASD diagnosis are shown in Table 10 in the Appendix.

**Cortical Thickness.** Table 3(a) reports the results of disease classification based on CT. As shown, CortiLife consistently outperforms all competing methods across datasets and tasks. It achieves improvements of 1.4%, 1.6%, and 0.2% in classification accuracy over the strongest baselines, respectively.

**Mean Curvature.** Table 3(b) presents the disease classification results based on MC. As shown, our proposed method achieves superior performance, reaching accuracies of 66.7%, 77.6%, and 93.9% on the CHD, ADHD, and ADNI datasets, respectively.

**Sulcal Depth.** Table 3(c) shows the results of disease classification based on SD. As shown, compared to the second-best model, CortiLife achieves an improvement of 1.8%, 0.7%, and 1.6% in accuracy, with showing the best AUCs for 79.9%, 65.7% and 99.1% in CHD, ADHD, and ADNI dataset, respectively.

The results demonstrate that our model achieves state-of-the-art performance across all lifespan stages and modalities, highlighting its superior generalization capability across both the entire lifespan and cortical modalities.

Table 4: Maximum Mean Discrepancy before vs. after patch-wise distribution encoding.

| Condition | Maximum Mean Discrepancy | | |
|---|---|---|---|
| | 1-3 (ys) vs 6-66 (ys) | 6-64 (ys) vs 55-95 (ys) | 1-3 (ys) vs 55-95 (ys) |
| Before scale-adaptive encoder | 0.852 | 0.592 | 1.029 |
| After scale-adaptive encoder | 0.473 | 0.271 | 0.933 |

Table 5: Results of ablation study in brain disorder diagnosis task.

| local | global | statistical | CT ACC | CT AUC | MC ACC | MC AUC | SD ACC | SD AUC |
|---|---|---|---|---|---|---|---|---|
| | | | | | **CHD** | | | |
| ✗ | ✔ | ✔ | 0.738±0.045 | 0.810±0.046 | 0.644±0.021 | 0.677±0.024 | 0.703±0.025 | 0.694±0.046 |
| ✔ | ✗ | ✔ | 0.740±0.012 | 0.798±0.017 | 0.569±0.012 | 0.610±0.024 | 0.718±0.034 | 0.740±0.021 |
| ✔ | ✔ | ✗ | 0.792±0.034 | 0.821±0.024 | 0.622±0.022 | 0.718±0.041 | 0.714±0.057 | 0.730±0.079 |
| ✔ | ✔ | ✔ | **0.806±0.011** | **0.823±0.026** | **0.667±0.031** | **0.776±0.014** | **0.739±0.033** | **0.799±0.073** |
| | | | | | **ADHD** | | | |
| ✗ | ✔ | ✔ | 0.633±0.031 | 0.661±0.010 | 0.617±0.011 | 0.604±0.022 | 0.598±0.032 | 0.595±0.019 |
| ✔ | ✗ | ✔ | 0.659±0.024 | 0.676±0.018 | 0.627±0.011 | 0.651±0.008 | 0.604±0.013 | 0.578±0.016 |
| ✔ | ✔ | ✗ | 0.615±0.008 | 0.620±0.017 | 0.589±0.021 | 0.621±0.017 | 0.588±0.032 | 0.585±0.045 |
| ✔ | ✔ | ✔ | **0.697±0.015** | **0.730±0.007** | **0.632±0.005** | **0.618±0.018** | **0.651±0.010** | **0.657±0.017** |
| | | | | | **AD** | | | |
| ✗ | ✔ | ✔ | 0.918±0.002 | 0.978±0.003 | 0.928±0.007 | 0.970±0.003 | 0.963±0.007 | 0.990±0.002 |
| ✔ | ✗ | ✔ | 0.925±0.002 | 0.975±0.003 | 0.918±0.020 | 0.972±0.004 | 0.955±0.012 | 0.977±0.004 |
| ✔ | ✔ | ✗ | 0.910±0.008 | 0.960±0.003 | 0.915±0.017 | 0.963±0.004 | 0.951±0.010 | 0.980±0.002 |
| ✔ | ✔ | ✔ | **0.928±0.002** | **0.981±0.001** | **0.939±0.011** | **0.973±0.002** | **0.972±0.002** | **0.991±0.002** |

## 4.5 VALIDATION OF PATCH-WISE DISTRIBUTION ENCODING

To further assess the scale-adaptive encoder, we compared feature discrepancies across age groups before and after applying it, using Maximum Mean Discrepancy (MMD) as a metric (Table 4). The consistently lower MMD values indicate reduced distributional shifts between age groups, showing that the encoder better aligns patch-wise feature distributions across the lifespan and yields more stable cortical representations. We also visualize the embedding distributions in Figure 5 in the Appendix. Finally, to verify that the benefits of scale-adaptive encoding are not simply due to adding statistical features, we replaced it with several alternative encoders and evaluated them on the ADHD classification task, with results reported in Section A.3.

## 4.6 ABLATION STUDY

We designed experiments to evaluate the effectiveness of different-level encoders within the proposed surface tokenizer. The evaluation was conducted under a pretraining + fine-tuning setting across classification tasks involving three cortical modalities and multiple age groups. The experimental setup and evaluation metrics follow the same settings as in Section 4.2. The results are presented in Table 5. Additionally, we observed that removing any encoder level leads to a performance decline, highlighting the effectiveness of each component. Notably, in the ADHD and ADNI classification tasks, excluding the statistical-level encoder resulted in a pronounced drop in accuracy. This can be attributed to the loss of regional statistical features, which impairs the model's ability to jointly capture heterogeneous cortical morphological characteristics, thereby degrading the quality of the learned representations.

## 5 CONCLUSION

We proposed CortiLife, the first unified framework for lifespan-consistent cortical surface modeling. By introducing a multi-level Surface Tokenizer, our approach addresses three central challenges in cortical analysis: complex geometry of cortical surfaces, registration-induced cortical homogenization and lifespan-related heterogeneity. Experimental evaluations across multiple datasets show that CortiLife consistently outperforms existing methods on brain disorder classification, age prediction and cortical surface parcellation. Most importantly, CortiLife provides a unified framework that exhibits strong generalization across the entire lifespan, diverse modalities, and multiple tasks.

ACKNOWLEDGMENTS

This research was supported by the National Natural Science Foundation of China(62476129) and the STI 2030-Major Projects (2022ZD0209000).

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

# A  APPENDIX

## A.1  CODE AVAILABILITY

Our codes are available in the link below.

https://github.com/xuepengcheng1231/CortiLife-A-Unified-Framework-for-Cortical-Representation-Learning-across-the-Lifespan.

## A.2  DATASETS

### A.2.1  DHCP

A total of 887 participants (478 males and 409 females) were enrolled in the Developing Human Connectome Project (DHCP), recruited at a single research site; the imaging hardware included Philips scanners, predominantly operating at a magnetic field strength of 3 T, with a representative model such as the Philips Achieva 3 T. For structural imaging, T2-weighted (T2w) and inversion-recovery T1-weighted (T1w) multi-slice Fast Spin-Echo (FSE) images were each acquired in sagittal and axial slice stacks, with an in-plane resolution of $0.80 \times 0.80$ mm$^2$ and 1.60 mm slices overlapped by 0.80 mm (except for T1w sagittal, which used a 0.74 mm overlap); key sequence parameters were as follows: T2w, TR/TE 12000/156 ms with SENSE factors 2.11 (axial) and 2.60 (sagittal); T1w (IR-FSE), TR/TI/TE 4795/1740/8.7 ms with SENSE factors 2.27 (axial) and 2.66 (sagittal). In addition, 3D MPRAGE images were acquired with a native resolution of approximately $0.80 \times 0.80 \times 0.80$ mm$^3$; typical parameters were TR/TI/TE 11/1400/4.6 ms with a SENSE factor of 1.2 (right–left). Regarding image processing, the official DHCP structural pipeline and surface reconstruction results were adopted in this study, with no additional processing performed; for additional details, please refer to `https://www.frontiersin.org/journals/neuroscience/articles/10.3389/fnins.2022.886772/full`.

### A.2.2  CHD

A total of 229 participants (123 males and 106 females) were enrolled in the Congenital Heart Disease (CHD) study, all recruited from a single research site. The imaging hardware consisted of Philips scanners, predominantly operating at a magnetic field strength of 3 T, with the Philips Ingenia 3 T as the representative model. For structural imaging, the 3D T1-TFE sequence was utilized, with a typical native resolution of approximately $0.50 \times 0.50 \times 0.50$ mm$^3$. The key sequence parameters were set as follows: TR 7.90 ms, TE 3.50 ms, and FA 8°. The image processing workflow was as follows: N3 intensity inhomogeneity correction, multi-atlas skull stripping, and tissue segmentation via multi-atlas label fusion; additionally, topological repair and reconstruction of the inner and outer cortical surfaces were performed. Subsequently, the spherical surfaces were aligned to the UNC 4D neonatal/infant cortical template, and surface parcellation was completed in accordance with the Desikan–Killiany atlas. Quality control (QC) checks were performed, and only data that passed QC were retained for subsequent analyses.

### A.2.3  CBCP

A total of 252 participants (133 males and 119 females) were enrolled in the Chinese baby connectome project(CBCP). The imaging hardware consisted of Philips scanners, predominantly operating at a magnetic field strength of 3 T, with the Philips Ingenia 3 T as the representative model. For structural imaging, the 3D T1-TFE sequence was utilized, with a typical native resolution of approximately $0.80 \times 0.80 \times 0.80$ mm$^3$. The key sequence parameters were set as follows: TR 6.50 ms, TE 2.30 ms, and FA 8°. The image processing workflow was as follows: N3 intensity inhomogeneity correction, multi-atlas skull stripping, and tissue segmentation via multi-atlas label fusion; additionally, topological repair and reconstruction of the inner and outer cortical surfaces were performed. Subsequently, the spherical surfaces were aligned to the UNC 4D neonatal/infant cortical template, and surface parcellation was completed in accordance with the Desikan–Killiany atlas. Quality control (QC) checks were performed, and only data that passed QC were retained for subsequent analyses.

### A.2.4 CCNP

A total of 559 participants (304 males and 255 females) were enrolled in the Chinese Color Nest Project(CCNP); the imaging hardware comprised Siemens scanners, predominantly operating at a magnetic field strength of 3 T with the Siemens Tim Trio 3 T as the representative model. For structural imaging, the MPRAGE sequence was utilized, with a typical native resolution of approximately $1.00 \times 1.00 \times 1.00$ mm$^3$. The key sequence parameters were set as follows: TR 2600 ms, TE 3.02 ms, TI 900 ms, and FA 8°. For image processing, cortical reconstruction and segmentation were performed using the `recon-all` pipeline of FreeSurfer 7 (FS7) with default parameters, and only data that passed quality control were retained for subsequent analyses; for additional details, please refer to `https://ccnp.scidb.cn/detail?dataSetId= c81f0e90a51b4cfca348ce4da6ca734e&version=V2&code=o00133`.

### A.2.5 ADHD-200

A total of 972 participants (599 males and 373 females) with Attention Deficit Hyperactivity Disorder-200 (ADHD-200 dataset) were enrolled in the study, recruited from 8 research sites; the imaging hardware included Siemens and Philips scanners, predominantly operating at a magnetic field strength of 1.5–3 T, with representative models such as the Siemens Tim Trio 3 T, Philips 3 T, and Siemens Avanto 1.5 T. For structural imaging, the MPRAGE sequence was utilized, with typical native resolutions of approximately $1.00 \times 1.00 \times 1.00$ mm$^3$ and $1.30 \times 1.00 \times 1.30$ mm$^3$, and key sequence parameters as follows: TR 2100–3500 ms, TE 2.95–3.70 ms, TI 900–1100 ms, and FA 7–10°. For image processing, cortical reconstruction and segmentation were performed using the `recon-all` pipeline of FreeSurfer 7 (FS7) with default parameters, and only data that passed quality control were retained for subsequent analyses; for additional details, please refer to `https://pubmed.ncbi.nlm.nih.gov/27423255/`.

### A.2.6 ABIDE II

A total of 1,114 participants (856 males and 258 females) were enrolled in the Autism Brain Imaging Data Exchange 2 (ABIDE II) study, recruited from 19 research sites; the imaging hardware included scanners from Philips, General Electric (GE), and Siemens, predominantly operating at a magnetic field strength of 1.5–3 T, with representative models such as the Siemens Tim Trio 3 T, GE MR750, and Philips Achieva 3 T. For structural imaging, the primary sequences utilized were MPRAGE, 3D FFE, and FSPGR, with typical native resolutions of approximately $1.00 \times 1.00 \times 1.30$ mm$^3$ and $1.30 \times 1.00 \times 1.30$ mm$^3$; the key sequence parameters ranged as follows: TR 2500–3000 ms, TE 2.30–8.30 ms, TI 853–1100 ms, and FA 7–10°. Regarding image processing, cortical reconstruction and segmentation were performed using the `recon-all` pipeline of FreeSurfer 7 (FS7) with default parameters, and only data that passed quality control were retained for subsequent analyses; for additional details, please refer to `https://www.nature.com/articles/mp201378`.

### A.2.7 ABIDE I

A total of 1,112 participants (948 males and 164 females) were enrolled in the Autism Brain Imaging Data Exchange 1 (ABIDE I) study, recruited from 20 research sites; the imaging hardware included scanners from Siemens, Philips, and General Electric (GE), predominantly operating at a magnetic field strength of 3 T, with representative models such as the Siemens Tim Trio 3 T, Siemens Allegra 3 T, and GE Signa. For structural imaging, the primary sequences utilized were MPRAGE, 3D FFE, and FSPGR, with typical native resolutions of approximately $1.00 \times 1.00 \times 1.00$ mm$^3$ and $1.00 \times 1.00 \times 1.20$ mm$^3$; the key sequence parameters ranged as follows: TR 1230–2530 ms, TE 1.73–4.60 ms, TI 624–1100 ms, and FA 7–10°. Regarding image processing, cortical reconstruction and segmentation were performed using the `recon-all` pipeline of FreeSurfer 7 (FS7) with default parameters, and only data that passed quality control were retained for subsequent analyses; for additional details, please refer to `https://www.nature.com/articles/mp201378`.

### A.2.8 HCP

A total of 1,206 participants (550 males and 656 females) were enrolled in the Human Connectome Project (HCP), recruited at a single research site; the imaging hardware included Siemens scanners, predominantly operating at a magnetic field strength of 3 T, with a representative model such as the

Table 6: Comparison of different distribution-encoding methods on ADHD-200 dataset.

| Setting | CT | | MC | | SD | |
|---|---|---|---|---|---|---|
| | ACC | AUC | ACC | AUC | ACC | AUC |
| Directly concatenation | 0.670±0.034 | 0.681±0.022 | 0.604±0.015 | 0.603±0.008 | 0.628±0.003 | 0.619±0.010 |
| BatchNorm+Linear | 0.645±0.049 | 0.664±0.024 | 0.597±0.021 | 0.595±0.057 | 0.591±0.039 | 0.603±0.037 |
| LayerNorm+Linear | 0.626±0.018 | 0.629±0.043 | 0.611±0.007 | 0.599±0.045 | 0.622±0.033 | 0.618±0.034 |
| Ours | **0.697±0.015** | **0.730±0.007** | **0.632±0.005** | **0.618±0.018** | **0.651±0.010** | **0.657±0.017** |

Siemens Skyra 3 T. The common native resolution was approximately $0.70\times0.70\times0.70$ mm$^3$, and the typical parameters were: TR 2400 ms, TE 2.14 ms, TI 1000 ms, and FA 8°. Regarding image processing, the Minimal Preprocessing Pipelines (HCP-MPP) provided by the HCP and FreeSurfer reconstruction results were adopted, with no additional processing performed on this basis; for additional details, please refer to `https://pubmed.ncbi.nlm.nih.gov/23668970/`.

### A.2.9 ADNI

A total of 7,597 participants (4,185 males and 3,412 females) were enrolled in the Alzheimer's Disease Neuroimaging Initiative (ADNI) study, recruited from 62 research sites; a mixed range of imaging devices was utilized, with representative models including 1.5–3 T scanners from Siemens, General Electric (GE), and Philips that are compatible with Magnetic Resonance Imaging (MRI). For structural imaging, the ADNI-specific MPRAGE sequence was the primary choice, with a typical native resolution of approximately $1.20\times1.20\times1.20$ mm$^3$. For image processing, cortical reconstruction and segmentation were performed using the `recon-all` pipeline of FreeSurfer 7 (FS7) with default parameters, and only data that passed quality control were retained for subsequent analyses; for additional details, please refer to `https://onlinelibrary.wiley.com/doi/full/10.1002/jmri.21049`.

### A.3 EFFECTIVENESS VALIDATION OF PATCH-WISE DISTRIBUTION ENCODING

#### A.3.1 COMPARISON WITH STATISTICAL-FEATURE INTEGRATION METHODS

To verify that the effectiveness of the scale-adaptive encoding does not simply arise from adding statistical features, we replaced the scale-adaptive encoder with some baselines and evaluated it on the ADHD classification task. The baselines include: 1) directly concatenation(i.e., concat[mean, std]); 2)LayerNorm(concat[mean, std]) with learnable weights(i.e., LayerNorm(concat[mean, std])+Linear); 3) BatchNorm(concat[mean, std]) with learnable weights(i.e., BatchNorm(concat[mean, std])+Linear). The results are shown in the Table 6. Our method still achieves the best performance. Notably, introducing LayerNorm leads to a clear degradation. The primary reason is that LayerNorm forces the statistics within each sample to lie on a similar scale, thereby suppressing the genuine distributional differences across samples and retaining only the relative within-sample relationships. In contrast, when performing weighting across multiple scales, our scale-adaptive encoder adopts an adaptive weighting scheme based on the original distribution. It treats each patch's own statistics as the principal component and uses statistics from other scales as auxiliary cues. This design preserves key distributional characteristics and avoids the issue where normalization operations inadvertently remove meaningful lifespan-relevant variability.

#### A.3.2 VISUALIZATION OF FEATURE DISTRIBUTIONS

We provide visual comparisons of the feature distributions before and after applying the scale-adaptive encoding, as shown in Figure 5. After incorporating the scale-adaptive encoder, the feature distributions across different age ranges and modalities become substantially more aligned, demonstrating that the proposed design effectively mitigates distribution shifts arising from both age and modality differences.

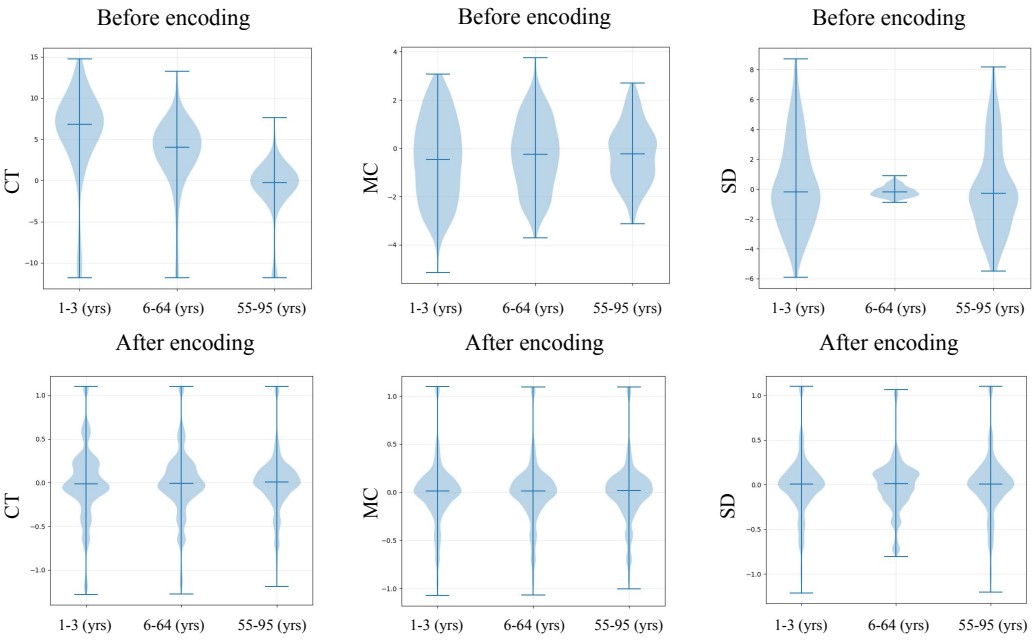

Figure 5: Feature distribution visualization before and after scale-adaptive encoding across age groups.

### A.4 EFFECTIVENESS VALIDATION OF TEACHER-STUDENT ARCHITECTURE

#### A.4.1 ANALYSIS OF MASK RATIO

We conducted the experiments to determine the mask ratio using the ADHD-200 dataset as an example. As shown in Table 7, the masking ratio has a clear influence on model performance across CT, MC, and SD. Among all the tested settings, the ratio of 0.75 consistently achieves the best results, yielding the highest ACC and AUC in CT (0.697/0.730) and SD (0.651/0.657), and the best ACC in MC (0.632). Although a ratio of 0.9 slightly improves the AUC for MC, it leads to a noticeable drop in ACC, indicating an imbalanced trade-off. Overall, 0.75 provides the most stable and superior performance across all modalities, confirming it as the optimal masking ratio for our model.

#### A.4.2 COMPARISON WITH RANDOM MASKING STRATEGY

To assess the contribution of the teacher-student architecture to representation learning, we conducted an ablation study in which the teacher-student mechanism was removed and replaced with a random-masking strategy. We performed experiments on the ADHD-200 dataset across three cortical modalities (CT, MC, and SD), and the results are presented in Table 8. As shown, removing the teacher-student architecture leads to a consistent degradation in accuracy and AUC across all modalities. Additionally, as the random mask ratio increases, the model performance on all metrics drops sharply, indicating that under high masking ratios a purely random strategy struggles to retain patches that are truly related to developmental semantics, which in turn leads to a clear degradation in the quality of the pre-trained representations. In contrast, when we adjust the mask ratio within the same range under the teacher-student architecture, although the performance shows some fluctuations, the overall decline is much smaller. This observation further suggests that, compared with relying only on random masking, the teacher-student architecture can more robustly learn high-quality representations aligned with developmental semantics under high masking conditions, thanks to the semantic guidance strategy introduced in our framework.

Table 7: Selection of mask ratio using ADHD-200 dataset as an example.

| Mask ratio | CT | | MC | | SD | |
|---|---|---|---|---|---|---|
| | ACC | AUC | ACC | AUC | ACC | AUC |
| 0.5 | 0.681±0.010 | 0.697±0.031 | 0.621±0.018 | 0.632±0.034 | 0.631±0.015 | 0.648±0.029 |
| 0.6 | 0.683±0.028 | 0.706±0.011 | 0.630±0.009 | 0.619±0.025 | 0.617±0.021 | 0.607±0.014 |
| 0.8 | 0.671±0.034 | 0.708±0.053 | 0.628±0.008 | 0.658±0.010 | 0.628±0.030 | 0.604±0.046 |
| 0.9 | 0.667±0.027 | 0.713±0.011 | 0.618±0.037 | **0.687±0.012** | 0.598±0.018 | 0.609±0.036 |
| 0.75(Ours) | **0.697±0.015** | **0.730±0.007** | **0.632±0.005** | 0.618±0.018 | **0.651±0.010** | **0.657±0.017** |

Table 8: Comparison of model performance with and without teacher-student architecture on ADHD-200 datset.

| Strategy | Ratio | CT | | MC | | SD | |
|---|---|---|---|---|---|---|---|
| | | ACC | AUC | ACC | AUC | ACC | AUC |
| Random Masking | 0.75 | 0.680±0.024 | 0.677±0.022 | 0.622±0.002 | 0.634±0.016 | 0.621±0.008 | 0.620±0.022 |
| | 0.8 | 0.655±0.036 | 0.643±0.028 | 0.624±0.023 | 0.572±0.057 | 0.630±0.021 | 0.588±0.009 |
| | 0.85 | 0.651±0.021 | 0.638±0.023 | 0.617±0.021 | 0.610±0.015 | 0.575±0.021 | 0.536±0.016 |
| | 0.9 | 0.631±0.009 | 0.630±0.035 | 0.619±0.018 | 0.614±0.021 | 0.527±0.071 | 0.568±0.016 |
| Teacher-student | 0.6 | 0.683±0.028 | 0.706±0.011 | 0.630±0.009 | 0.619±0.025 | 0.617±0.021 | 0.607±0.014 |
| | 0.8 | 0.671±0.034 | 0.708±0.053 | 0.628±0.008 | 0.658±0.010 | 0.628±0.030 | 0.604±0.046 |
| | 0.9 | 0.667±0.027 | 0.713±0.011 | 0.618±0.037 | **0.687±0.012** | 0.598±0.018 | 0.609±0.036 |
| | 0.75 | **0.697±0.015** | **0.730±0.007** | **0.632±0.005** | 0.618±0.018 | **0.651±0.010** | **0.657±0.017** |

### A.4.3 VISUALIZATION OF REGIONS FROM MASKED SELF-DISTILLATION FRAMEWORK

We visualize the visible patches at different age stages selected by the masked self-distillation framework. The results are shown in the Figure 6. The visualization demonstrates that, in infancy and early childhood, key brain developmental regions mainly appear in the superior frontal gyrus, precentral, paracentral, and other motor-related brain regions, as well as some visual regions including the lateral occipital; in adolescence, the core regions that are mainly selected are distributed in the prefrontal and temporal lobes; in adulthood, the main regions show a distribution pattern similar to that in adolescence; and in the elderly stage, we do not observe relatively concentrated regions corresponding to aging, which suggests that there may already exist a global aging phenomenon in elderly individuals. The above visualization results indicate that teacher-guided masking indeed focuses more on cortical regions that are closely related to developmental semantics.

### A.5 FEW-SHOT PERFORMANCE EVALUATION ON BRAIN DISORDER DIAGNOSIS

To assess few-shot performance, we trained CortiLife with only 20% and 40% of the training data, reserving the rest for testing. As shown in Table 9, CortiLife outperforms baselines even with limited data. For example, on CHD with mean curvature features, it achieved 65.1% accuracy using 40% of the data, surpassing all baselines; on ADNI with sulcal depth, it reached 95.4%, nearly matching the second-best model (95.6%). These results highlight the strong generalization ability of CortiLife under data-scarce conditions.

Table 9: Evaluation of few-shot learning performance of CortiLife on brain disease classification tasks.

| Methods | Training Percentage | CT | | MC | | SD | |
|---|---|---|---|---|---|---|---|
| | | ACC | AUC | ACC | AUC | ACC | AUC |
| | | | | CHD | | | |
| CortiLife | 0.2 | 0.626±0.005 | 0.602±0.023 | 0.631±0.036 | 0.632±0.040 | 0.631±0.019 | 0.683±0.016 |
| | 0.4 | 0.701±0.008 | 0.693±0.018 | 0.651±0.010 | 0.703±0.011 | 0.654±0.023 | 0.702±0.036 |
| | 0.8 | 0.806±0.011 | 0.823±0.026 | 0.667±0.031 | 0.776±0.014 | 0.739±0.033 | 0.799±0.073 |
| | | | | ADHD | | | |
| CortiLife | 0.2 | 0.601±0.007 | 0.606±0.002 | 0.608±0.001 | 0.561±0.006 | 0.604±0.004 | 0.605±0.002 |
| | 0.4 | 0.641±0.013 | 0.667±0.008 | 0.623±0.010 | 0.628±0.009 | 0.618±0.015 | 0.611±0.018 |
| | 0.8 | 0.697±0.015 | 0.730±0.007 | 0.632±0.005 | 0.618±0.018 | 0.651±0.010 | 0.657±0.017 |
| | | | | AD | | | |
| CortiLife | 0.2 | 0.856±0.004 | 0.931±0.001 | 0.881±0.002 | 0.947±0.001 | 0.86±0.003 | 0.926±0.001 |
| | 0.4 | 0.894±0.002 | 0.961±0.004 | 0.924±0.010 | 0.971±0.002 | 0.954±0.001 | 0.988±0.002 |
| | 0.8 | 0.928±0.002 | 0.981±0.001 | 0.939±0.011 | 0.973±0.002 | 0.972±0.002 | 0.991±0.002 |

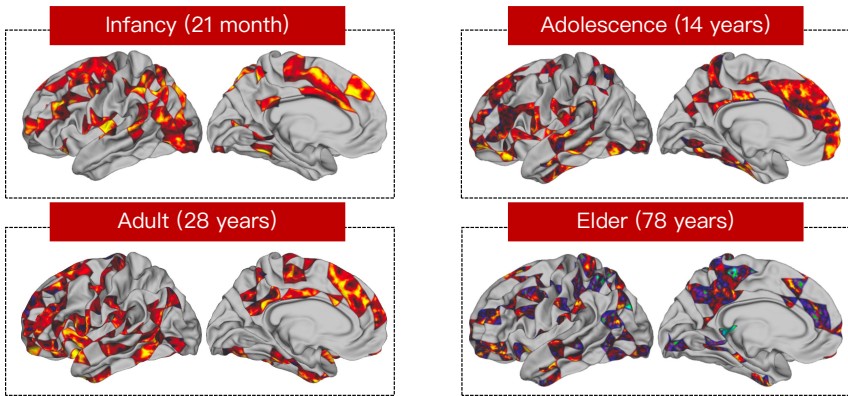

Figure 6: This image illustrates the visible patches selected by masked self-distillation framework. The process selectively eliminates less relevant tokens, retaining only those with strong semantic significance.

Table 10: Results of brain disorder diagnosis on ABIDE I dataset.

| Method | ASD | | | | | |
| --- | --- | --- | --- | --- | --- | --- |
| | CT | | MC | | SD | |
| | ACC | AUC | ACC | AUC | ACC | AUC |
| SphericalCNN | 0.563±0.016 | 0.582±0.020 | 0.572±0.008 | 0.572±0.007 | 0.574±0.022 | 0.564±0.035 |
| Spherical U-Net | 0.561±0.009 | 0.553±0.009 | 0.564±0.003 | 0.573±0.034 | 0.571±0.009 | 0.577±0.004 |
| WSSADN | 0.566±0.021 | 0.573±0.021 | 0.564±0.010 | 0.547±0.026 | 0.560±0.025 | 0.567±0.005 |
| NeuroExplainer | 0.558±0.019 | 0.577±0.012 | 0.571±0.007 | 0.574±0.016 | 0.554±0.007 | 0.567±0.002 |
| SurfaceVisionTransformer | 0.552±0.027 | 0.547±0.024 | 0.561±0.011 | 0.542±0.031 | 0.542±0.032 | 0.528±0.017 |
| CLIP | 0.572±0.002 | 0.570±0.017 | 0.573±0.007 | 0.571±0.003 | 0.593±0.007 | 0.592±0.011 |
| ACLIP | 0.569±0.027 | 0.575±0.021 | 0.573±0.015 | 0.570±0.008 | 0.569±0.005 | 0.557±0.028 |
| DetailCLIP | 0.574±0.012 | 0.581±0.016 | 0.541±0.012 | 0.539±0.017 | 0.604±0.022 | 0.604±0.027 |
| CARZero | 0.577±0.003 | **0.584±0.017** | 0.568±0.005 | 0.568±0.012 | 0.599±0.015 | 0.597±0.008 |
| CortiLife | **0.587±0.004** | 0.574±0.009 | **0.584±0.006** | **0.576±0.015** | **0.623±0.010** | **0.636±0.025** |

## A.6 COMPARISON WITH SOTA METHODS ON ABIDE I DATASET

Table 10 summarizes the ASD classification performance on the ABIDE I dataset across three cortical modalities (CT, MC, and SD). Overall, CortiLife achieves the most competitive results among all comparison methods, demonstrating consistently strong performance across both ACC and AUC metrics.

For the CT modality, CortiLife attains the highest accuracy (0.587±0.004), outperforming all baselines, while maintaining a comparable AUC to the best-performing model. For the MC modality, CortiLife achieves the best accuracy (0.584±0.006) and also records one of the strongest AUC values (0.576±0.015), indicating that the learned representations capture more discriminative curvature-related patterns associated with ASD. Notably, on the SD modality, CortiLife delivers the best performance on both metrics, with an accuracy of 0.623±0.010 and an AUC of 0.636±0.025, significantly surpassing all competing approaches. These results highlight the robustness and generalization ability of CortiLife across heterogeneous cortical attributes.

We further conducted classification experiments on three age ranges in ABIDE I: 6-18 years, 18-30 years, and 30-55 years, with the results shown in the Table 11. We found that our model still exhibited consistent and strong performance across all age-stratified stages. In the 6-18, 18-30, and 30-55 age ranges, the ACC reached up to 0.612, 0.694, and 0.778, respectively, and the AUC reached up to 0.608, 0.768, and 0.773, respectively. It is worth noting that some age ranges include relatively few subjects, so performance on these stratified subsets may not fully reflect the overall accuracy on the full dataset. Even under this data limited setting, CortiLife still shows consistently superior performance across all age ranges, indicating strong generalization ability within different age groups.

Table 11: Results of brain disorder diagnosis on age-stratified ABIDE I dataset.

| Method | 6-18 years | | 18-30 years | | 30-55 years | |
|---|---|---|---|---|---|---|
| | ACC | AUC | ACC | AUC | ACC | AUC |
| SphericalCNN | 0.593±0.014 | 0.603±0.019 | 0.629±0.057 | 0.631±0.047 | 0.742±0.073 | 0.583±0.169 |
| Spherical U-NET | 0.588±0.017 | 0.576±0.002 | 0.649±0.015 | 0.591±0.027 | 0.656±0.014 | 0.629±0.147 |
| WSSADN | 0.593±0.008 | 0.579±0.014 | 0.657±0.052 | 0.664±0.062 | 0.701±0.043 | 0.673±0.058 |
| NeuroExplainer | 0.593±0.015 | 0.602±0.023 | 0.638±0.039 | 0.661±0.045 | 0.704±0.088 | 0.699±0.041 |
| SurfaceVisionTransformer | 0.577±0.025 | 0.580±0.005 | 0.646±0.041 | 0.602±0.077 | 0.721±0.078 | 0.679±0.117 |
| CLIP | 0.586±0.016 | 0.581±0.028 | 0.639±0.063 | 0.601±0.027 | 0.711±0.038 | 0.728±0.085 |
| ACLIP | 0.586±0.043 | 0.589±0.023 | 0.652±0.064 | 0.640±0.076 | 0.758±0.045 | 0.684±0.114 |
| DetailCLIP | 0.601±0.012 | 0.604±0.034 | 0.659±0.042 | 0.707±0.017 | 0.757±0.040 | 0.730±0.095 |
| CARZero | 0.602±0.018 | 0.606±0.012 | 0.662±0.033 | 0.712±0.052 | 0.744±0.043 | 0.728±0.083 |
| CortiLife | **0.612±0.008** | **0.608±0.011** | **0.694±0.032** | **0.768±0.043** | **0.778±0.039** | **0.773±0.098** |

Table 12: Comparison of model performance with and without metadata language prompting on ADHD-200 dataset.

| Setting | CT | | MC | | SD | |
|---|---|---|---|---|---|---|
| | ACC | AUC | ACC | AUC | ACC | AUC |
| w/o $L_{CLIP}$ | 0.630±0.005 | 0.603±0.008 | 0.609±0.013 | 0.585±0.011 | 0.605±0.013 | 0.566±0.028 |
| w/ $L_{CLIP}$ | **0.697±0.015** | **0.730±0.007** | **0.632±0.005** | **0.618±0.018** | **0.651±0.010** | **0.657±0.017** |

## A.7 EFFECTIVENESS VALIDATION OF METADATA LANGUAGE PROMPTING

We conducted the experiments to validate the effectiveness of the $L_{CLIP}$ loss using the ADHD-200 dataset as an example. We removed this loss term(i.e., disabling metadata language prompting) to examine its role in guiding the vision encoder to learn lifespan-aware representations. As shown in the Table 12, eliminating $L_{CLIP}$ leads to substantial drops across all evaluation metrics, underscoring the critical importance of metadata language prompting for achieving semantically aligned and developmentally informative cortical embeddings.

Additionally, we conducted the experiments to validate the effectiveness of prompts(i.e., age and sex) using the ADHD-200 dataset as an example. As shown in Table 13, removing either the age prompt or the sex prompt leads to consistent drops in accuracy and AUC across all three modalities. The degradation is more pronounced when the age prompt is removed, further underscoring its critical role in enabling the model to learn lifespan-aware information. In comparison, the sex prompt has a smaller yet complementary effect, providing additional developmental cues that further refine the learned representations.

Table 13: Results of disease classification on ADHD-200 dataset in different architecture.

| Scale-adaptive encoder | Age prompting | Sex prompting | CT | | MC | | SD | |
|---|---|---|---|---|---|---|---|---|
| | | | ACC | AUC | ACC | AUC | ACC | AUC |
| ✗ | ✗ | ✔ | 0.604±0.005 | 0.621±0.002 | 0.595±0.015 | 0.606±0.016 | 0.591±0.007 | 0.602±0.002 |
| ✔ | ✗ | ✔ | 0.660±0.013 | 0.679±0.014 | 0.604±0.010 | 0.605±0.008 | 0.614±0.029 | 0.565±0.024 |
| ✔ | ✔ | ✗ | 0.687±0.005 | 0.703±0.006 | 0.621±0.019 | **0.662±0.021** | 0.629±0.036 | 0.635±0.047 |
| ✔ | ✔ | ✔ | **0.697±0.015** | **0.730±0.007** | **0.632±0.005** | 0.618±0.018 | **0.651±0.010** | **0.657±0.017** |

