# OpenReview forum: "CortiLife: A Unified Framework for Cortical Representation Learning across the Lifespan"
_ICLR.cc/2026/Conference — ICLR 2026 Poster_

### Official Review · Reviewer_Hj8o · 2025-10-28

**Soundness:** 2
**Presentation:** 2
**Contribution:** 3
**Rating:** 4
**Confidence:** 4

**Summary:**

This paper addresses cortical representation learning across the lifespan, collecting an impressive dataset spanning 26 weeks to 95 years (13,928 subjects) and proposing CortiLife with multi-level surface tokenization. The experimental scope is comprehensive and results show competitive performance.

However, I have significant concerns about the claimed contribution. The paper positions itself as "the first lifespan-aware framework," but the evidence primarily demonstrates strong performance from CLIP trained on diverse age data rather than architectural innovation for cross-age generalization. Critical issues include: (1) Figure 4's circular reasoning (age/gender embeddings reflect explicit text inputs, not learned capability), (2) unclear whether baselines used the same pretraining data (unfair comparison?), (3) no direct evidence that scale-adaptive encoding addresses distribution shift vs. adding features, and (4) evaluation gaps (no adults 30-55, no per-age-group breakdown). The paper needs to clarify what specifically makes this lifespan-aware beyond data aggregation and provide ablations isolating architectural contributions. With revisions addressing these fundamental questions, this could become a solid contribution. Currently, the novelty and validity of claims are insufficient.

**Strengths:**

- Significant data collection effort (9 datasets, 13,928 subjects across lifespan)
- Comprehensive experimental scope (multiple tasks, modalities, settings)
- Extensive baseline comparisons
- Generally competitive empirical performance
- Technically sound surface patchification approach

**Weaknesses:**

**W1. Unclear Contribution Beyond SOTA Performance**
The paper claims to be "the first unified vision-language framework for lifespan-aware cortical representation learning," but the actual technical novelty separating this from standard multi-dataset training is unclear. The key question remains: Is this truly lifespan-aware architecture innovation, or simply a CLIP-based model trained on diverse age datasets? The performance gaps over baselines are often marginal (e.g., Table 3a: 0.806 vs 0.792 for CHD), raising concerns about whether the architectural components specifically enable lifespan generalization or if aggregate data is the primary factor.

**W2. Circular Reasoning in Figure 4**
Figure 4 visualizes embeddings separated by age, gender, and diagnosis - but these exact attributes are explicitly provided as text inputs via PubMedBERT (Section 2.3: "The age of the subject is [age]. The gender of the subject is [gender]..."). This is not evidence of learned representation quality but rather a trivial reflection of the input. The paper presents this as validation of the model's capability, which is misleading.

**W3. Distribution Shift Solution Lacks Validation**
The paper identifies distribution shifts across lifespan as a key challenge (Figure 1c) and proposes scale-adaptive encoding (Eq. 1-2) as the solution. However:

- The ablation study (Table 4) only shows that removing "statistical" encoding reduces performance
- No evidence that this specifically addresses distribution shift vs. simply adding more features
- No analysis showing that embeddings from different age groups are actually aligned in the representation space

**W4. Inconsistent Experimental Setup**

- Baseline models' pretraining data is not specified - did they use the same 8 datasets?
- Fine-tuning only evaluates 3 specific age groups (infancy, adolescence, elderly), missing adult ages 30-55
- Table 1 shows HCP covers 22-36 years, but it's only used for zero-shot evaluation, not fine-tuning
- The claim of "across the lifespan" is not systematically validated across all age ranges

**W5. Poor Presentation Quality**

- Related Work relegated to Appendix without justification (highly unusual)
- Inconsistent terminology: "ADHD" (disease) vs "ADNI" (dataset) mixed in Table 3 headers
- Acronyms (CHD, ADHD, ADNI) undefined until Appendix A.3, first appearing in main text without explanation
- Figure 2 is overly complex and difficult to parse

**W6. Unmotivated and Underexplored Teacher-Student Design**
The paper employs masked self-distillation with teacher-student framework (Section 2.2), masking 75% of patches based on attention scores. However:

- **No ablation study** comparing with/without this design (Table 4 doesn't include it)
- **No computational efficiency metrics** reported (training time, inference speed, memory)
- **No justification for 75% masking ratio** - why not 50%, 60%, or 80%? This appears borrowed from MAE without domain-specific validation
- The claimed motivation ("information redundancy") is not validated
- Appears to be standard MAE-style masking without cortex-specific justification

This is concerning because:

1. If efficiency is the goal → no speedup metrics provided
2. If better representation is the goal → no ablation showing improvement
3. The 75% ratio may be suboptimal for cortical surfaces where spatial anatomical structure matters

**Questions:**

**Q1.** What specific architectural components enable lifespan generalization beyond simply training on multi-age datasets? Can you provide ablation studies showing performance when age information is removed from text prompts?

**Q2.** Were baseline methods (CLIP, ACLIP, DetailCLIP, CARZero) pretrained on the same 8 datasets as CortiLife? If not, how is this a fair comparison?

**Q3.** Figure 4 shows embeddings clustered by attributes that were explicitly input. How does this demonstrate learned capability rather than input reflection? Can you show embeddings on data where age/gender were NOT provided as text input?

**Q4.** How does the scale-adaptive encoder (Eq. 1-2) specifically address distribution shift? Can you visualize embedding distributions across age groups to show alignment?

**Q5.** Why is fine-tuning only performed on 3 datasets spanning limited age ranges? What about performance on young/middle-aged adults (30-55 years)?

**Q6.** Table 3 shows many baseline methods with comparable performance. For example, in Table 3c (SD), WSSADN achieves 0.935 vs your 0.972 on ADNI - but WSSADN had no pretraining. How much gain is from pretraining vs. lifespan-aware architecture?

**Q7.** In zero-shot evaluation (Table 2), CARZero performs catastrophically poorly (DICE ~0.01). Was this implementation correct? This seems like an implementation error.

**Q8.** What is the specific purpose of the teacher-student masked self-distillation framework? The paper employs this design masking 75% of patches, but the motivation is unclear. Is this for computational efficiency (if so, where are the speedup metrics?), or for learning better representations (if so, where is the ablation comparing single-model vs. teacher-student)? Table 4's ablation study does not include this comparison. The paper cites "information redundancy" in cortical data, but provides no evidence that this design specifically addresses it. This appears to be standard MAE-style masking - what makes this cortex-specific?

**Q9.** Where is the computational cost analysis? If the student processes only 25% of patches during training, what is the actual speedup in training time, inference time, and memory consumption compared to processing all patches? Does this enable scaling to higher-resolution cortical meshes? Without efficiency metrics, the teacher-student design appears unmotivated - if there's no computational benefit and no representation quality benefit (per Q8), why use it?

**Q10.** Why specifically 75% masking ratio? Section 2.2 states "Patches with the lowest 75% attention values are masked out," but no justification or ablation study for this choice is provided. Was this borrowed from MAE (He et al., 2022) which used 75% for natural images? Cortical surfaces have fundamentally different properties - high spatial autocorrelation, fixed anatomical structure (motor cortex, visual cortex locations), and critical geometric features. Shouldn't the masking ratio be validated for this domain? Can you provide ablations testing 50%, 60%, 80%, or 90% masking to show 75% is optimal for cortical surfaces?

---

> ### Author Response · Authors · 2025-11-22
> **Thank the reviewer for the constructive suggestions. We have provided detailed responses to all comments.**
>
> # Response to Reviewer 3
> We appreciate your positive comments on our work, including “Comprehensive experimental scope” and “Extensive baseline comparisons”. Detailed responses to all comments are provided below.
> ---
> ## (1)W1&Q1&Q6 Model’s performance gains come from multi-center data training or from the architectural design.
> Thanks for the questions.
>
> First, we would like to clarify that our method is not a simple application of CLIP to datasets from different age groups. Instead, our framework is a carefully designed CLIP-style VLM architecture specifically tailored to address the core challenges of lifespan cortical representation learning. These challenges arise at three levels.
>
> 1) Cortical surfaces lie on a non-Euclidean manifold, making standard vision-language models designed for Euclidean image grids inapplicable. To bridge this mismatch, we propose a Surface Tokenizer that converts the complex cortical manifold into a structured sequence representation compatible with VLM architectures.
>
> 2) Cortical morphology is highly homogeneous across subjects because of the registration, making it difficult to learn subject-specific features. To enhance discriminability, we introduce a multi-level encoding mechanism that captures patch-level geometric and morphological features across multiple spatial granularity, enabling the model to become more sensitive to subtle structural variations.
>
> 3) The most critical challenge is the distribution shift across developmental stages. Prior studies have shown that cortical morphology follows dynamic lifespan trajectories, and different structural modalities exhibit heterogeneous developmental patterns (i.e., different levels of distribution drift), greatly increasing the difficulty of designing a unified representation model that generalizes across ages and modalities. To address this, we design Distribution Encoding (i.e., the scale-adaptive encoder) into the Surface Tokenizer, projecting each patch into multiple shared statistical scale spaces and fusing them via a data-dependent weighting mechanism. This design enables the model to adaptively counteract lifespan-related distribution shifts across modalities. In addition, we introduce Lifespan-Aware Prompting, which uses metadata prompts to explicitly guide the model toward learning developmental-stage-specific structural semantics. Together, these components enable our framework to perform lifespan-aware cortical representation learning, rather than merely pretraining on multi-age datasets without architectural innovations.

---

> > ### Author Response · Authors · 2025-11-22
> >
> > ## (1)W1&Q1&Q6 Model’s performance gains come from multi-center data training or from the architectural design.
> > Second, to further validate the effectiveness of our lifespan-aware module designs, we conducted an additional ablation study by removing both the Scale-Adaptive Encoder and the age information in the prompting module. We evaluated this ablation on the ADHD dataset under the classification setting. As shown in Table below, removing either component leads to a clear degradation in classification performance, demonstrating that these lifespan-aware modules play an essential role in enabling the model to learn robust representations across developmental stages.
> >
> > | Scale-adaptive encoder | Age prompting | CT ACC        | CT AUC        | MC ACC        | MC AUC        | SD ACC        | SD AUC        |
> > |:------------------------|:---------------:|:---------------:|:---------------:|:---------------:|:---------------:|:---------------:|:---------------:|
> > | ×                      | ×             | 0.604±0.005   | 0.621±0.002   | 0.595±0.015   | 0.606±0.016   | 0.591±0.007   | 0.602±0.002   |
> > | √                      | ×             | 0.660±0.013   | 0.679±0.014   | 0.604±0.010   | 0.605±0.008   | 0.614±0.029   | 0.565±0.024   |
> > | √                      | √             | **0.697±0.015** | **0.730±0.007** | **0.632±0.005** | **0.618±0.018** | **0.651±0.010** | **0.657±0.017** |
> >
> > Moreover, our experimental results also demonstrate that our method is not a trivial application of CLIP. The four CLIP-based SOTA baselines we compare against (CLIP, ACLIP, DetailCLIP, and CARZero) use exactly the same text prompt design as ours and are all trained on the same multiple datasets from different age groups. Under the fair comparison setting, our method achieves consistently superior performance across all evaluation tasks, including more accurate age prediction (Table 2(a)), improved cortical parcellation results (Table 2(b)), and more accurate disease classification performance (Table 3). These results indicate that the performance gains stem from our architectural innovations rather than differences in training data or prompt configuration.
> >
> > Regarding the reviewer’s concern that the performance improvements over baselines appear modest (e.g., 0.806 vs. 0.792 on CHD in Table 3a), we would like to clarify that no existing SOTA method is able to perform well across all tasks. Most competing models achieve strong results only on specific datasets or tasks, but their performance degrades significantly when evaluated more broadly. For instance, while WSSADN shows comparable accuracy to CortiLife on three tasks (CHD&CT, AD&CT, and ADHD&MC), its performance drops substantially on all other classification tasks. This clearly indicates that traditional models are task-specialized and lack the robustness required for diverse downstream applications. In contrast, our goal is to build a generalizable lifespan-aware cortical representation model, rather than optimizing for a single dataset. CortiLife leverages large-scale pretraining to learn a unified representation space that transfers consistently across different tasks, age ranges, and diagnostic categories. The fact that CortiLife delivers strong, and often superior, performance across all evaluation tasks demonstrates the importance and effectiveness of building a general-purpose representation model. Thus, even if the improvement on a single dataset may appear modest, the broad, stable, and consistent gains observed across the entire benchmark clearly show the value and necessity of our design.
> >
> > ## (2) W2 & Q3 Circular Reasoning in Figure 4
> >
> > Thanks. To clarify, metadata such as age and sex is introduced only during pretraining through PubMedBERT. Its purpose is to guide semantic alignment between the vision and text modalities, enabling the vision encoder to implicitly encode demographic-related variations. During inference, however, all metadata inputs are removed, and the downstream tasks are performed using cortical morphology alone. Thus, the embeddings visualized in Figure 4 are generated solely from cortical surface data, without providing age, sex, or any other metadata to the vision encoder. Thus, the visualization is not a reflection of the textual prompts. We have revised the manuscript to describe this process more explicitly.

---

> > > ### Author Response · Authors · 2025-11-22
> > >
> > > ## (3) W3 & Q4 Distribution Shift Solution via scale-adaptive encoder
> > >
> > > Thanks. Below, we will clarify how the scale-adaptive encoder addresses distribution shift from both theoretical and empirical perspectives.
> > >
> > > First, from the theoretical standpoint, the distribution shift in cortical representation learning primarily arises from lifespan-related anatomical development, which causes substantial heterogeneity in feature distributions across ages (as shown in Figure 1(c)). The core idea of our scale-adaptive encoder is to introduce a set of scale bases, each corresponding to a different statistical level of the feature distribution. Patch representations from all ages are projected onto these shared scale bases and then fused adaptively based on their original statistics. This design simultaneously unifies cross-age distribution levels and preserves age-specific distribution characteristics (see Figure 2 and Figure 5). The detailed formulation is as follows: 1) For each age group, we compute the mean and standard deviation for every patch and project them into $n$ scale spaces via Eq. (1), yielding multi-scale representations $z(x_m)$. Each scale corresponds to a distinct distribution magnitude; 2) Based on the original statistical characteristics of each age group, we compute the adaptive weights for different scales using Eq. (2). Scales that are more compatible with the patch’s intrinsic distribution are assigned larger weights, enabling the encoder to better preserve the age-specific statistical patterns. The final patch embedding is the weighted sum of multi-scale representations, allowing each patch to be mapped to the mixture of statistical scales rather than being forced into a single normalized distribution. It is worth emphasizing that, compared with conventional normalization techniques, our method exhibits substantially greater robustness in handling outliers. Even for individual samples in which certain cortical regions show pronounced deviations, our mechanism can still project these regions into the same set of multi-scale statistical spaces. This prevents the modeling failures or representational distortions that commonly arise when traditional normalization amplifies extreme numerical deviations.
> > >
> > > Second, from the experimental standpoint, we conducted two additional experiments to verify whether the proposed scale-adaptive encoder can effectively mitigate distribution shift. (1) In the first experiment, we compared the feature discrepancies across different age groups before and after applying the scale-adaptive encoder. We quantitatively evaluated the distribution shift using Maximum Mean Discrepancy (MMD) (Table 4 in revised manuscript) and further provided distribution visualizations for qualitative analysis (Figure 6 in revised manuscript). (2) In the second experiment, we compared a naïve baseline that directly concatenates statistical features (mean and standard deviation) with our scale-adaptive encoder on the ADHD classification task. Across all results, the scale-adaptive encoder consistently demonstrates clear advantages, confirming its effectiveness in alleviating cross-age distribution heterogeneity.
> > >
> > > 1) Feature comparison before and after applying the scale-adaptive encoder
> > >
> > > Using the CT modality as an example, we computed the MMD between different age groups before and after applying the scale-adaptive encoder. The results are shown in the table below. As can be observed, the distributional discrepancies across age groups are substantially reduced after applying the scale-adaptive encoder. In addition, we provide visual comparisons of the feature distributions before and after the transformation in Figure 6 of the revised manuscript appendix. The distribution plots further demonstrate that the proposed scale-adaptive encoding markedly decreases the feature discrepancies across different age groups.
> > >
> > > | Condition                     | 1–3 (ys) vs 6–64 (ys) | 6–64 (ys) vs 55–95 (ys) | 1–3 (ys) vs 55–95 (ys) |
> > > |:----------------------------:|:----------------------:|:------------------------:|:-----------------------:|
> > > | **Maximum Mean Discrepancy** |                        |                          |                         |
> > > | Before scale-adaptive encoder | 0.852                  | 0.592                    | 1.029                   |
> > > | After scale-adaptive encoder  | 0.473                  | 0.271                    | 0.933                   |

---

> > > > ### Author Response · Authors · 2025-11-22
> > > >
> > > > 2) Comparison with directly adding statistical features
> > > >
> > > > To further verify that the effectiveness of the scale-adaptive encoding does not simply arise from adding statistical features, we replaced the scale-adaptive encoder with a baseline that directly concatenates patch-level statistical features (mean and standard deviation) to the original inputs, and evaluated it on the ADHD classification task. The results are shown in the table below. As demonstrated, directly adding these statistical features leads to substantially worse performance compared to the proposed scale-adaptive encoding, demonstrating the effectiveness of the scale-adaptive encoding.
> > > >
> > > > | Method                   | CT ACC          | CT AUC          | MC ACC          | MC AUC          | SD ACC          | SD AUC          |
> > > > |:--------------------------|:-----------------:|:-----------------:|:-----------------:|:-----------------:|:-----------------:|:-----------------:|
> > > > | Directly Concatenation   | 0.670 ± 0.034   | 0.681 ± 0.022   | 0.604 ± 0.015   | 0.603 ± 0.008   | 0.628 ± 0.003   | 0.619 ± 0.010   |
> > > > | Scale-adaptive encoding  | **0.697 ± 0.015** | **0.730 ± 0.007** | **0.632 ± 0.005** | **0.618 ± 0.018** | **0.651 ± 0.010** | **0.657 ± 0.017** |
> > > >
> > > > ## (4) W4 & Q2 Were baseline methods (CLIP, ACLIP, DetailCLIP, CARZero) pretrained on the same 8 datasets as CortiLife? If not, how is this a fair comparison?：
> > > >
> > > > For all baseline methods that require pretraining (CLIP, ACLIP, DetailCLIP, and CARZero), we adopt exactly the same training pipeline and experimental settings as CortiLife to ensure fairness. Specifically: 1) all models are pretrained on the same set of eight datasets; 2) each method is evaluated on the diverse downstream tasks using an identical experimental settings and evaluation configuration. We have clarified these details explicitly in the revised version of the manuscript.
> > > >
> > > > ## (5) W4&Q5 Inconsistent Experimental Setup：
> > > >
> > > > 1) Table 1 shows HCP covers 22-36 years, but it's only used for zero-shot evaluation, not fine-tuning
> > > >
> > > >     Our fine-tuning tasks are disease-classification tasks, which require both patient and healthy control samples. Since the HCP dataset contains only healthy subjects, it is not suitable for this type of task.
> > > >
> > > > 2) Fine-tuning only evaluates 3 specific age groups (infancy, adolescence, elderly), missing adult ages 30-55.
> > > >
> > > >     The reason we did not include a fine-tuning task specifically for the 30-55 age range is that none of our currently collected datasets contain subjects exclusively within this age interval. To address this concern, we conducted an additional experiment on the ABIDE I dataset (including 539 ASD subjects and 573 healthy subjects), whose subject ages span 6-64 years and therefore cover the 30-55 age group. We followed the same fine-tuning protocol used in our other downstream tasks, and the results are shown in the table below. As demonstrated, CortiLife continues to outperform other SOTA methods within this age range. These results have been added to the appendix of the revised manuscript.
> > > >
> > > > | Method                    | CT ACC        | CT AUC        | MC ACC        | MC AUC        | SD ACC        | SD AUC        |
> > > > |---------------------------|---------------|---------------|---------------|---------------|---------------|---------------|
> > > > | SphericalCNN              | 0.563±0.016   | 0.582±0.020   | 0.572±0.008   | 0.572±0.007   | 0.574±0.022   | 0.564±0.035   |
> > > > | SphericalUNet             | 0.561±0.009   | 0.553±0.009   | 0.564±0.003   | 0.573±0.034   | 0.571±0.009   | 0.577±0.004   |
> > > > | WSSADN                    | 0.566±0.021   | 0.573±0.021   | 0.564±0.010   | 0.547±0.026   | 0.560±0.025   | 0.567±0.005   |
> > > > | NeuroExplainer            | 0.558±0.019   | 0.577±0.012   | 0.571±0.007   | 0.574±0.016   | 0.554±0.007   | 0.567±0.002   |
> > > > | SurfaceVisionTransformer  | 0.552±0.027   | 0.547±0.024   | 0.561±0.011   | 0.542±0.031   | 0.542±0.032   | 0.528±0.017   |
> > > > | CLIP                      | 0.572±0.002   | 0.570±0.017   | 0.573±0.007   | 0.571±0.003   | 0.593±0.007   | 0.592±0.011   |
> > > > | ACLIP                     | 0.569±0.027   | 0.575±0.021   | 0.573±0.015   | 0.570±0.008   | 0.569±0.005   | 0.557±0.028   |
> > > > | DetailCLIP                | 0.574±0.012   | 0.581±0.016   | 0.541±0.012   | 0.539±0.017   | 0.604±0.022   | 0.604±0.027   |
> > > > | CARZero                   | 0.577±0.003   | **0.584±0.017** | 0.568±0.005   | 0.568±0.012   | 0.599±0.015   | 0.597±0.008   |
> > > > | CortiLife                 | **0.587±0.004** | 0.574±0.009   | **0.584±0.006** | **0.576±0.015** | **0.623±0.010** | **0.636±0.025** |

---

> > > > > ### Author Response · Authors · 2025-11-22
> > > > >
> > > > > ## (6) W5 Poor Presentation Quality:
> > > > >
> > > > > 1) Related Work relegated to Appendix without justification.
> > > > >
> > > > >     Due to the page limitations of the ICLR main paper, we followed the practice of several prior ICLR works (e.g., Labram, Jiang et al., ICLR 2024) and moved the Related Work section to the appendix. In the revised version, we have updated the manuscript accordingly and moved the Related Work section back into the main text.
> > > > >
> > > > > 2) Inconsistent terminology and Acronyms (CHD, ADHD, ADNI) undefined.
> > > > >
> > > > >     We have revised the manuscript to correct this error.
> > > > >
> > > > > 3) Figure 2 is overly complex and difficult to parse.
> > > > >
> > > > >     Figure 2(a) presents the overall framework of our model, which consists of the surface tokenizer and the CLIP-based VLM. Figure 2(b) illustrates the construction pipeline of the surface tokenizer, including the three encoding modules: Local Topology Encoding (Fig. 2(b1)), Global Interaction Encoding (Fig. 2(b2)), and Patch-wise Distribution Encoding (Fig. 2(b3)). Following your suggestions, we have revised Figure 2 to make these three components more explicit and easier to understand. In the Local Topology Encoding module, we emphasize how local topological features within each patch are extracted. In the Global Interaction Encoding module, we clarify how interaction relationships are established across patches and how global contextual dependencies are captured. In the Patch-wise Distribution Encoding module, we highlight the effect of the scale-adaptive encoder and illustrate how this module mitigates cross-age statistical scale shifts. These refinements make the figure more intuitive and help readers better understand the motivation and functionality of each encoding component.
> > > > >
> > > > > ## (7) W6 & Q8 & Q9 Unmotivated and Underexplored Teacher-Student Design
> > > > >
> > > > > Thanks. As described in the paper, the teacher-student masked self-distillation framework is designed to guide the model toward learning higher-quality representations, rather than to improve computational efficiency. Below, we clarify how this design contributes to representation learning from both an architectural perspective and empirical evidence.
> > > > >
> > > > > From an architectural perspective, we adopt the teacher-student masked self-distillation framework because cortical surface data exhibit substantial spatial redundancy: cortical maturation exhibits well-documented spatial co-development[1][2][3], where exhibits changes in one region often covary with changes in other regions. If the model processes all patches simultaneously, it may be dominated by these repeated local signals, which in turn weakens its ability to capture developmentally meaningful and discriminative structural variations. However, purely random masking is not suitable for cortical data, because it provides no guarantee that developmentally critical cortical regions will be preserved, which will prevent the model to learn from patches with effective and rich semantics. To address this limitation, we use a teacher-student architecture. The teacher network receives all cortical patches and computes attention maps to evaluate the semantic importance of each patch, thereby identifying the most informative cortical regions. These selected key patches are then fed into the student network, which performs reconstruction and representation learning under the semantic guidance of the teacher. This design enables the model to concentrate on developmentally informative cortical regions and substantially enhances the quality of the learned cortical representations.
> > > > >
> > > > > From the experimental perspective, we added an ablation study in which the teacher-student mechanism was replaced with pure random masking. The results are shown below. Compared with random masking, the teacher-student framework achieves substantially higher classification accuracy in all tasks, demonstrating the effectiveness of incorporating this design. These results have been added to Table 10 in the Appendix of the revised manuscript.
> > > > >
> > > > > | Setting           | CT ACC        | CT AUC        | MC ACC        | MC AUC        | SD ACC        | SD AUC        |
> > > > > |:-------------------|:---------------:|:---------------:|:---------------:|:---------------:|:---------------:|:---------------:|
> > > > > | random masking    | 0.680±0.024   | 0.677±0.022   | 0.622±0.002   | 0.634±0.016   | 0.621±0.008   | 0.620±0.022   |
> > > > > | Teacher-student   | **0.697±0.015** | **0.730±0.007** | **0.632±0.005** | 0.618±0.018   | **0.651±0.010** | **0.657±0.017** |
> > > > >
> > > > > [1]Alexander-Bloch A, Giedd J N, Bullmore E. Imaging structural co-variance between human brain regions[J]. Nature reviews neuroscience, 2013, 14(5): 322-336.
> > > > >
> > > > > [2]Mechelli A, Friston K J, Frackowiak R S, et al. Structural covariance in the human cortex[J]. Journal of Neuroscience, 2005, 25(36): 8303-8310.
> > > > >
> > > > > [3]Zielinski B A, Gennatas E D, Zhou J, et al. Network-level structural covariance in the developing brain[J]. Proceedings of the National Academy of Sciences, 2010, 107(42): 18191-18196.

---

> > > > > > ### Author Response · Authors · 2025-11-22
> > > > > >
> > > > > > ## (8) W6 & Q10 Ablation study of masking ratio
> > > > > >
> > > > > > We selected a 75% masking ratio by following prior studies such as TARDRL (MICCAI 2025) (1) and BrainX (CIKM 2025) (2), both of which report that a 75% masking ratio yields the best performance for medical imaging tasks. In addition, following the reviewer’s suggestion, we further evaluated masking ratios of 50%, 60%, 80%, and 90% by retraining the model and comparing their performance to the 75% setting on the ADHD classification task. The results (shown in the table below) indicate that the 75% ratio achieves the best performance in most cases, confirming the suitability and robustness of this choice. These results have been added to Table 7 in the Appendix of the revised manuscript.
> > > > > >
> > > > > >
> > > > > > | Mask ratio   | CT ACC        | CT AUC        | MC ACC        | MC AUC        | SD ACC        | SD AUC        |
> > > > > > |--------------|---------------|---------------|---------------|---------------|---------------|---------------|
> > > > > > | 0.5          | 0.681±0.010   | 0.697±0.031   | 0.621±0.018   | 0.632±0.034   | 0.631±0.015   | 0.648±0.029   |
> > > > > > | 0.6          | 0.683±0.028   | 0.706±0.011   | 0.630±0.009   | 0.619±0.025   | 0.617±0.021   | 0.607±0.014   |
> > > > > > | 0.8          | 0.671±0.034   | 0.708±0.053   | 0.628±0.008   | 0.658±0.010   | 0.628±0.030   | 0.604±0.046   |
> > > > > > | 0.9          | 0.667±0.027   | 0.713±0.011   | 0.618±0.037   | **0.687±0.012** | 0.598±0.018   | 0.609±0.036   |
> > > > > > | 0.75 (Ours)  | **0.697±0.015** | **0.730±0.007** | **0.632±0.005** | 0.618±0.018   | **0.651±0.010** | **0.657±0.017** |
> > > > > >
> > > > > > [1] Zhao Y, et al. TARDRL: Task-Aware Reconstruction for Dynamic Representation Learning of fMRI[C]//International Conference on Medical Image Computing and Computer-Assisted Intervention. Cham: Springer Nature Switzerland, 2024: 700-710.
> > > > > >
> > > > > > [2] Cui Z,et al. BrainX: A Universal Brain Decoding Framework with Feature Disentanglement and Neuro-Geometric Representation Learning[C]//Proceedings of the 34th ACM International Conference on Information and Knowledge Management. 2025: 478-487.
> > > > > >
> > > > > > ## (9) Q7 In zero-shot evaluation (Table 2), CARZero performs catastrophically poorly (DICE ~0.01). Was this implementation correct? This seems like an implementation error.
> > > > > >
> > > > > > The markedly low DICE scores of CARZero reported in Table 2 can be largely attributed to a mismatch between its cross-modal alignment strategy and the semantic granularity required for cortical parcellation. CARZero was originally designed for chest X-ray disease classification, a domain in which radiology reports routinely provide localized textual descriptions tied to specific anatomical regions. Consequently, the model’s alignment paradigm is optimized to associate local visual patches with spatially localized textual semantics.
> > > > > >
> > > > > > In our pretraining setting, however, the textual prompts encode only global, subject-level attributes (e.g., age, sex), without any spatial specificity. Such global semantics cannot be mapped to individual cortical patches. As a result, during cross-attention alignment, the global textual signal becomes dominant, driving patch representations toward a homogeneous global embedding and suppressing the local structural variability that parcellation critically relies upon.
> > > > > >
> > > > > > Because successful parcellation depends on distinguishing subtle, region-specific morphological differences, this collapse of patch-level representations substantially undermines CARZero’s ability to delineate cortical areas, leading to the extremely low DICE scores observed. By contrast, CLIP-based baselines (CLIP, ACLIP, DetailCLIP) adopt a global vision-text alignment paradigm and therefore avoid the local-global semantic mismatch, resulting in significantly better performance, although still behind our proposed method.

---

> ### Comment · Reviewer_Hj8o · 2025-11-25
> **First Response from Reviewer 3 #1**
>
> Thank you for the detailed responses and additional experiments. While I appreciate the effort, several fundamental concerns remain unaddressed. Below are my consolidated questions:
>
> Q1. Circular Reasoning in Figure 4 - Age Information Leakage
> I understand that metadata is removed during inference, but this misses the point of my original concern. During training, you explicitly provide age information via text prompts (e.g., "The age of the subject is 65"). The vision encoder is therefore directly supervised to encode age, making Figure 4's age-based clustering an expected consequence of this supervision rather than evidence of learned developmental understanding.
> To demonstrate that your model truly learns age-aware representations (rather than simply memorizing explicit age labels), please provide:
>
> Unseen age interpolation: Show t-SNE embeddings for age ranges not present in any training dataset (e.g., 35-45 years, which fall between your CCNP and ABIDE cohorts). Do these unseen ages smoothly interpolate between known age groups, or do they collapse to the nearest training distribution?
> Age-blind training ablation: Train a version of CortiLife WITHOUT age information in text prompts, then visualize whether embeddings still exhibit age-related structure. This would demonstrate that the model learns age-related patterns from cortical morphology itself, not from text supervision.
>
> Without these experiments, Figure 4 only validates that your model can follow explicit text instructions, not that it captures intrinsic developmental trajectories.
>
> Q2. Statistical Encoding vs. Multi-Level Architecture
> Your ablation study (Table 4) reveals an interesting pattern:
>
> local | global | statistical | CHD CT ACC | Gain over baseline
>   ×   |   ✓    |      ✓      |   0.738    | -
>   ✓   |   ×    |      ✓      |   0.740    | +0.2%p
>   ✓   |   ✓    |      ×      |   0.792    | +5.4%p
>   ✓   |   ✓    |      ✓      |   0.806    | +1.4%p (over statistical-only)
>
> This shows that statistical encoding contributes 5.4%p, while local and global encodings combined add only 1.4%p on top. This suggests that scale-adaptive encoding is doing most of the heavy lifting, with local/global encodings providing marginal benefits.
> Questions:
>
> Why do local and global encodings contribute so little? If registration-induced homogenization is a major challenge (as claimed), shouldn't multi-scale geometric features (local + global) be more critical?
> Have you compared against simpler baselines that use statistical features without the scale-adaptive mechanism? For example:
>
> LayerNorm(concat[mean, std]) without learnable scale weights
> Simply appending mean/std as additional input channels
>
>
>
> This would isolate whether the gains come from the scale-adaptive weighting mechanism specifically, or just from adding statistical features in general.
>
> Q3. Teacher-Student Masking Motivation
> Your motivation states: "cortical maturation exhibits spatial co-development... If the model processes all patches simultaneously, it may be dominated by these repeated local signals."
> However, this reasoning seems inconsistent with standard self-supervised learning principles:
>
> If regions co-develop (exhibit redundancy), then random masking should be MORE effective, because unmasked regions can provide sufficient signal to reconstruct masked ones via spatial correlations.
> Teacher-guided selection based on attention scores introduces inductive bias—why should we assume that teacher attention corresponds to "developmentally critical" regions? Where is the neuroanatomical validation?
>
> Your ablation shows:
> Random masking:     0.680
> Teacher-student:    0.697  (+1.7%p)
> Questions:
>
> Did you experiment with higher random masking ratios (e.g., 80%, 85%, 90%) to see if random masking can match teacher-student performance with more aggressive masking?
> Can you visualize which patches the teacher selects and verify they correspond to known developmentally-critical regions (e.g., prefrontal cortex, cingulate gyrus)?
> Have you compared against standard MAE-style random masking with the same 75% ratio? The current "random masking" baseline may be a weaker implementation.
>
> Given the modest 1.7%p improvement and added architectural complexity, the teacher-student design needs stronger justification.
>
> Q4. Age-Stratified Performance for ABIDE I
> Thank you for adding ABIDE I results. However, the current evaluation doesn't validate performance in the 30-55 age range specifically, since ABIDE I spans 6-64 years and likely has an uneven age distribution.
> Request: Please provide age-stratified performance breakdown:
>
> 6-18 years
> 18-30 years
> 30-55 years
> 55-64 years
>
> This would directly demonstrate whether CortiLife maintains consistent performance across the "missing" adult age range, or if performance degrades in under-represented age bins.

---

> > ### Author Response · Authors · 2025-11-29
> >
> > # Response to Reviewer 3
> >
> > ## Q1 Circular Reasoning in Figure 4
> >
> > Thanks. First, we would like to clarify that the setting of our method is not the same as the paradigm assumed by the reviewer. The goal of CortiLife is not to automatically generalize from limited age ranges to unseen age ranges under partially missing age labels, but to make use of large-scale, cross-age data and learn cortical developmental representations covering the entire lifespan by specifically designing some lifespan-aware architectures. In other words, the design philosophy of CortiLife is to allow the model to “see” brains from as many age ranges as possible during training, rather than requiring the model to infer unseen age ranges from limited observed ones. Under this setting, the age prompt is not a form of “information leakage,” but an intentional supervision signal for constructing a lifespan-aware joint representation space.
> >
> > Second, the purpose of Figure 4 is to examine whether, after the explicit age-supervised training, the model can still naturally exhibit age-related structure in the representation space on a completely unseen dataset when age information is removed at inference and only cortical morphology is provided. On the unseen AD dataset, the embeddings naturally form well-separated clusters aligned with age groups, indicating that the model has learned a stable internal mapping from cortical morphology to age-related semantics. Thus, what might be interpreted as “remembering age labels” is in fact the model learning a representation function that organizes cortical morphology along a continuous developmental axis. Figure 4 therefore reflects the generalization of this learned function to unseen data, rather than mere memorization of training labels.
> >
> > Based on the above motivation, we believe the two experiments proposed by the reviewer are not necessary in our framework, for the following reasons:
> >
> > 1.	Regarding the “interpolation validation on unseen age intervals”:
> >
> >     Our model does not rely on the ability to extrapolate from the distribution of some age ranges to unseen age ranges, so there is no theoretical requirement to validate such “interpolation ability”.
> > 2.	Regarding the “ablation experiment removing age textual prompts”:
> >
> >     The assumption raised by the reviewer, namely “the model should still automatically learn age-related patterns without age prompts”, corresponds to an unsupervised structural developmental modeling task. However, CortiLife adopts explicit age semantic supervision to construct a lifespan-aware representation space, so this ablation experiment is not applicable to our method. In addition, we would like to further emphasize that, due to the high inter-individual variability of the human brain, automatically extracting cortical developmental patterns purely through unsupervised learning is a highly challenging open problem that deserves further in-depth investigation in the future, but it is beyond the scope of this work.

---

> > ### Author Response · Authors · 2025-11-29
> >
> > ## Q2 Roles of scale-adaptive weighting mechanism
> >
> > Thanks. Regarding your inference about the contribution rates of different encoding channels, we would like to clarify that our ablation experiments are conducted by removing one module at a time and observing the resulting change in downstream classification performance, rather than using a hierarchical or accumulative ablation setting. As shown in the table, removing the local encoding leads to a 6.8%p drop in classification accuracy (0.806→0.738), removing the global encoding leads to a 6.6%p drop (0.806→0.740), and removing the statistical encoding leads to a 1.4%p drop (0.806→0.792). Thus, the relative contributions of the three encoding modules follow local encoding > global encoding >> statistical encoding. This pattern is consistent with our theoretical analysis. We will make the design of the ablation experiments clearer in the paper to avoid similar misunderstandings.
> >
> > | local | global | statistical | CHD CT ACC | Decrease under baseline |
> > |-------|--------|-------------|-------------|--------------------------|
> > | ×     | √      | √           | 0.738       | -6.8%p                  |
> > | √     | ×      | √           | 0.740       | -6.6%p                  |
> > | √     | √      | ×           | 0.792       | -1.4%p                  |
> > | √     | √      | √           | 0.806       | -----                   |
> >
> > To rule out the possibility that the performance gain comes “merely from adding statistical features,” we conducted more experiments on the ADHD-200 dataset. As shown in the table below, our method still achieves the best performance. Notably, introducing LayerNorm leads to a clear degradation. The primary reason is that LayerNorm forces the statistics within each sample to lie on a similar scale, thereby suppressing the genuine distributional differences across samples and retaining only the relative within-sample relationships. In contrast, when performing weighting across multiple scales, our scale-adaptive encoder adopts an adaptive weighting scheme based on the original distribution. It treats each patch’s own statistics as the principal component and uses statistics from other scales as auxiliary cues. This design preserves key distributional characteristics and avoids the issue where normalization operations inadvertently remove meaningful lifespan-relevant variability.
> >
> > | Setting               | CT ACC          | CT AUC          | MC ACC          | MC AUC          | SD ACC          | SD AUC          |
> > |-----------------------|-----------------|-----------------|-----------------|-----------------|-----------------|-----------------|
> > | Directly concatenation| 0.670±0.034   | 0.681±0.022   | 0.604±0.015   | 0.603±0.008   | 0.628±0.003   | 0.619±0.010   |
> > | BatchNorm+Linear      | 0.645±0.049     | 0.664±0.024     | 0.597±0.021     | 0.595±0.057     | 0.591±0.039     | 0.603±0.037     |
> > | LayerNorm+Linear      | 0.626±0.018     | 0.629±0.043     | 0.611±0.007     | 0.599±0.045     | 0.622±0.033     | 0.618±0.034     |
> > | Ours                  | **0.697±0.015** | **0.730±0.007** | **0.632±0.005** | **0.618±0.018** | **0.651±0.010** | **0.657±0.017** |

---

> > ### Author Response · Authors · 2025-11-29
> >
> > ## Q3. Teacher-Student Masking Motivation
> >
> > Thanks. Here, we further clarify the motivation for adopting the teacher-student masking architecture. We agree that when the data contain spatial redundancy (e.g., co-development), random masking can effectively leverage spatial correlations under a reconstruction-only objective, as the unmasked patches are sufficient to predict the masked ones. However, our framework jointly optimizes two objectives: reconstruction (via MAE) and developmental semantic alignment (via CLIP). For semantic alignment, it is crucial that the visible patches include regions rich in meaningful developmental information. Under a high masking ratio, purely random masking cannot guarantee this, and key semantic regions may be removed, preventing the model from learning representations that capture essential developmental semantics. To address this, we use the teacher’s attention maps to introduce inductive bias and preferentially select high-attention patches. Representations learned from these regions are more effective for CLIP-style alignment with text prompts. On top of this, we employ a reconstruction loss to ensure that fine-grained information in masked patches can still be recovered.
> >
> > Second, regarding why teacher attention better highlights development-related regions, in the teacher branch, we introduce a [CLS] token to represent global brain semantics. Under the joint supervision of the CLIP loss and the teacher-student consistency loss, this [CLS] token is explicitly optimized to encode high-level developmental information such as age, sex, and health status. We then use this development-aware [CLS] as the Query and treat each patch as the Key/Value to compute an attention distribution, retaining patches that receive high attention from [CLS], i.e., regions most strongly related to the global developmental semantic representation. Figure 7 in the revised manuscript visualizes the visible patches across age stages. In infancy, key brain developmental regions mainly appear in the superior frontal gyrus, precentral and other motor-related regions, as well as visual regions including the lateral occipital cortex. In adolescence, the core regions are primarily distributed in the prefrontal and temporal lobes. In adulthood, the main regions exhibit a distribution pattern similar to adolescence. In the elderly stage, we no longer observe relatively concentrated regions, suggesting a more global aging phenomenon. These visualizations indicate that teacher-guided masking indeed focuses more on cortical regions that are closely related to developmental semantics.
> >
> > In terms of experiments, the random masking results we previously reported were already obtained with a 75% masking ratio. Following the reviewer’s suggestion, we further evaluated random masking with 80%, 85%, and 90% masking ratios on the ADHD task, as shown in the table below. As the random mask ratio increases, the model performance on all metrics drops sharply, indicating that under high masking ratios a purely random strategy struggles to retain patches that are truly related to developmental semantics, which in turn leads to a clear degradation in the quality of the pre-trained representations. In contrast, when we adjust the mask ratio within the same range under the teacher-student architecture, although the performance shows some fluctuations, the overall decline is much smaller. This observation further suggests that, compared with relying only on random masking, the teacher-student architecture can more robustly learn high-quality representations aligned with developmental semantics under high masking conditions, thanks to the semantic guidance strategy introduced in our framework.
> >
> > | Strategy        | Ratio | CT ACC        | CT AUC        | MC ACC        | MC AUC        | SD ACC        | SD AUC        |
> > |-----------------|-------|---------------|---------------|---------------|---------------|---------------|---------------|
> > | Random Masking  | 0.75  | 0.680±0.024   | 0.677±0.022   | 0.622±0.002   | 0.634±0.016   | 0.621±0.008   | 0.620±0.022   |
> > | Random Masking  | 0.8   | 0.655±0.036   | 0.643±0.028   | 0.624±0.023   | 0.572±0.057   | 0.630±0.021   | 0.588±0.009   |
> > | Random Masking  | 0.85  | 0.651±0.021   | 0.638±0.023   | 0.617±0.021   | 0.610±0.015   | 0.575±0.021   | 0.536±0.016   |
> > | Random Masking  | 0.9   | 0.631±0.009   | 0.630±0.035   | 0.619±0.018   | 0.614±0.021   | 0.527±0.071   | 0.568±0.016   |
> > | Teacher-student | 0.6   | 0.683±0.028   | 0.706±0.011   | 0.630±0.009   | 0.619±0.025   | 0.617±0.021   | 0.607±0.014   |
> > | Teacher-student | 0.8   | 0.671±0.034   | 0.708±0.053   | 0.628±0.008   | 0.658±0.010   | 0.628±0.030   | 0.604±0.046   |
> > | Teacher-student | 0.9   | 0.667±0.027   | 0.713±0.011   | 0.618±0.037   | **0.687±0.012** | 0.598±0.018   | 0.609±0.036   |
> > | Teacher-student | 0.75  | **0.697±0.015** | **0.730±0.007** | **0.632±0.005** | 0.618±0.018   | **0.651±0.010** | **0.657±0.017** |

---

> > ### Author Response · Authors · 2025-11-29
> >
> > ## Q4. Age-Stratified Performance for ABIDE I
> >
> > We did not initially conduct age-stratified experiments on the ABIDE I dataset mainly because its age distribution is highly imbalanced. Specifically, the numbers of ASD and healthy control (HC) subjects in each age range are as follows: 6-18 years: 351 / 359, 18-30 years: 109 / 130, 30-55 years: 43 / 35, and 55-64 years: 1 / 3.
> >
> > According to the reviewer’s suggestion, we further conducted classification experiments on three age ranges in ABIDE I: 6-18 years, 18-30 years, and 30-55 years, with the results shown as below. We found that our model exhibited very consistent and strong performance across all age-stratified stages. In the 6-18, 18-30, and 30-55 age ranges, the ACC reached up to 0.612, 0.694, and 0.778, respectively, and the AUC reached up to 0.608, 0.768, and 0.773, respectively. It is worth noting that some age ranges include relatively few subjects, so performance on these stratified subsets may not fully reflect the overall accuracy on the full dataset. Even under this data limited setting, CortiLife still shows consistently superior performance across all age ranges, indicating strong generalization ability within different age groups.
> >
> > | Method                    | 6–18y ACC      | 6–18y AUC      | 18–30y ACC      | 18–30y AUC      | 30–55y ACC      | 30–55y AUC      |
> > |---------------------------|----------------|----------------|------------------|------------------|------------------|------------------|
> > | SphericalCNN              | 0.593±0.014    | 0.603±0.019    | 0.629±0.057      | 0.631±0.047      | 0.742±0.073      | 0.583±0.169      |
> > | SphericalUNET             | 0.588±0.017    | 0.576±0.002    | 0.649±0.015      | 0.591±0.027      | 0.656±0.014      | 0.629±0.147      |
> > | WSSADN                    | 0.593±0.008    | 0.579±0.014    | 0.657±0.052      | 0.664±0.062      | 0.701±0.043      | 0.673±0.058      |
> > | NeuroExplainer            | 0.593±0.015    | 0.602±0.023    | 0.638±0.039      | 0.661±0.045      | 0.704±0.088      | 0.699±0.041      |
> > | SurfaceVisionTransformer  | 0.577±0.025    | 0.580±0.005    | 0.646±0.041      | 0.602±0.077      | 0.721±0.078      | 0.679±0.117      |
> > | CLIP                      | 0.586±0.016    | 0.581±0.028    | 0.639±0.063      | 0.601±0.027      | 0.711±0.038      | 0.728±0.085      |
> > | ACLIP                     | 0.586±0.043    | 0.589±0.023    | 0.652±0.064      | 0.640±0.076      | 0.758±0.045      | 0.684±0.114      |
> > | DetailCLIP                | 0.601±0.012    | 0.604±0.034    | 0.659±0.042      | 0.707±0.017      | 0.757±0.040      | 0.730±0.095      |
> > | CARZero                   | 0.602±0.018    | 0.606±0.012    | 0.662±0.033      | 0.712±0.052      | 0.744±0.043      | 0.728±0.083      |
> > | CortiLife                 | **0.612±0.008** | **0.608±0.011** | **0.694±0.032**  | **0.768±0.043**  | **0.778±0.039**  | **0.773±0.098**  |

---

> ### Comment · Reviewer_Hj8o · 2025-11-25
> **First Comment from Reviewer 3 # 2**
>
> Q5. Distribution Shift Resolution - Elderly Subjects
> Your MMD results show:
>                     1-3 vs 6-64  |  6-64 vs 55-95  |  1-3 vs 55-95
> Before:                0.852     |     0.592       |     1.029
> After:                 0.473     |     0.271       |     0.933
> Reduction:             44.5%     |     54.2%       |     9.3%
> While early-to-middle age transitions show substantial improvement, the elderly range (1-3 vs 55-95) shows minimal reduction (1.029 → 0.933, only 9.3%). This suggests that scale-adaptive encoding is less effective for the largest distributional gap.
> Questions:
>
> Why does the method struggle with elderly subjects specifically? Is this due to limited elderly data in pretraining, or a fundamental limitation of the scale-adaptive mechanism?
> Can you compare your unified model against age-specific models trained separately on each age group (infancy, adolescence, adult, elderly)? If scale-adaptive encoding truly resolves distribution shift, the unified model should match or exceed age-specific models.
>
>
> Q6. Direct Concatenation Baseline Weakness
> Your comparison shows:
> Direct concatenation:      0.670
> Scale-adaptive encoding:   0.697  (+2.7%p)
> The direct concatenation baseline seems insufficiently competitive. The 2.7%p gap is not "substantially worse" as claimed—it's marginal.
> Request: Please compare against intermediate baselines:
>
> LayerNorm(concat[mean, std]) + Linear (normalization without scale-adaptive weights)
> Standard batch normalization applied to mean/std features
> Multi-scale features WITHOUT adaptive weighting (i.e., fixed uniform weights)
>
> This would clarify whether the gains come from the scale-adaptive weighting mechanism specifically, or simply from normalizing statistical features.
>
> Q7. CARZero Implementation Verification
> You explain CARZero's failure (DICE ~0.01) as due to "local-global semantic mismatch." However:
>
> If CARZero is fundamentally incompatible with this task, why include it as a baseline? This raises concerns about either implementation error or cherry-picking baselines.
> CLIP, ACLIP, and DetailCLIP also use global text prompts, yet they perform reasonably well on parcellation. Why does only CARZero fail catastrophically?
>
> Request:
>
> Verify your CARZero implementation by reproducing results from their original paper (on chest X-ray tasks)
> If CARZero is indeed incompatible, consider removing it from Table 2 or clearly marking it as "not applicable" rather than presenting a misleading performance comparison
>
>
> Summary
> My core concerns remain:
>
> Figure 4 demonstrates memorization of explicit text labels, not learned age-aware representations (needs unseen age interpolation test)
> Scale-adaptive encoding dominates contribution; local/global encodings are marginal (needs simpler statistical feature baselines)
> Teacher-student motivation contradicts SSL principles (co-development should favor random masking; needs neuroanatomical validation)
> Distribution shift resolution is weakest for elderly subjects, the largest gap (needs comparison against age-specific models)
>
> I appreciate the additional experiments, but these fundamental questions must be addressed to support the claim that CortiLife is "the first lifespan-aware framework" rather than primarily a CLIP model trained on diverse age data with statistical feature augmentation.

---

> > ### Author Response · Authors · 2025-11-29
> >
> > ## Q5. Distribution Shift Resolution
> >
> > Regarding the relatively smaller reduction in the distribution discrepancy between ages 1-3 and 55-95, we further clarify that this is caused by the weighting mechanism of the scale-adaptive encoding itself. Specifically, this mechanism performs weighted fusion over multiple statistical scales, and its weighting rule has the property that “the closer to its own statistical characteristics, the higher the weight”. Specifically, for each scale, if a patch’s original mean/std is closest to that scale, then this scale will receive a higher adaptive weight, while the weights of other scales will decay progressively. Therefore, for age ranges such as 1-3 and 55-95 whose original statistical distributions are far apart (as shown in Figure 8 of the revised manuscript), the patches in each range rely more on their own “nearest” scale during encoding, and the alignment effect across more distant scales is relatively limited, which leads to a slightly smaller extent of alleviation of the distribution discrepancy between these two extreme age ranges. Nevertheless, our method still significantly reduces the distribution difference between these two age ranges and achieves more evident improvements over other age spans.
> >
> > In response to the reviewer’s suggestion of training separate models on each age group and making comparisons, we address this from two perspectives. First, our comparative experiments already include a strong supervised non pretrained baseline that directly trains a classification model on the corresponding downstream dataset within a specific age stage, and the results show that our CortiLife consistently outperforms these task specific supervised models. Second, we further add an age range matched pretraining experiment. Using the ADHD task as an example, we select a pretraining dataset whose age range exactly matches that of ADHD-200, from 7 to 27 years, perform pretraining only on data within this range, and then fine tune on ADHD-200 to obtain the classification results. As reported in the table below, our method still achieves superior performance in this matched setting.
> >
> > | Method      | CT ACC        | CT AUC        | MC ACC        | MC AUC        | SD ACC        | SD AUC        |
> > |-------------|---------------|---------------|---------------|---------------|---------------|---------------|
> > | CLIP        | 0.686±0.005   | 0.680±0.011   | 0.626±0.003   | 0.614±0.003   | 0.609±0.009   | 0.622±0.011   |
> > | ACLIP       | 0.664±0.019   | 0.676±0.004   | 0.598±0.013   | 0.532±0.015   | 0.614±0.015   | 0.581±0.008   |
> > | DetailCLIP  | 0.656±0.013   | 0.657±0.015   | 0.624±0.008   | 0.615±0.021   | 0.642±0.007   | 0.624±0.021   |
> > | CAEZero     | 0.669±0.003   | 0.695±0.005   | 0.612±0.003   | 0.580±0.018   | 0.630±0.009   | 0.631±0.004   |
> > | CortiLife   | **0.697±0.015** | **0.730±0.007** | **0.632±0.005** | **0.618±0.018** | **0.651±0.010** | **0.657±0.017** |

---

> > ### Author Response · Authors · 2025-11-29
> >
> > ## Q6. Direct Concatenation Baseline Weakness Your comparison
> >
> > Thanks. We added two baseline methods, where we normalize the mean and standard deviation with LayerNorm or BatchNorm and then feed them into a Linear layer to enhance their representational capacity. As shown in the table below, these intermediate baselines still do not surpass our scale adaptive encoding and even drop sharply. This is because, after applying LayerNorm or BatchNorm, the original patch statistics lose critical cross region and cross subject differences. For example, LayerNorm forces the features within each subject onto similar scales, which erases inter subject distribution differences and leads to a marked drop in classification performance, even lower than directly concatenating the features. This evidence further shows that the performance improvement does not simply come from normalizing or linearly mapping the statistical features, but instead arises from the proposed scale adaptive weighting mechanism for multi scale feature modeling and fusion.
> >
> > In addition, it should be emphasized that ADHD classification based on cortical structure is a highly challenging task, and in this context, an improvement of 2.7% is not negligible but a performance gain of practical significance. For example, in papers published at MICCAI 2025, the results in [1] show that the accuracy improvement on ADHD is only 0.5%, and the accuracy improvement in [2] is only 1.26%. Therefore, the 2.7% improvement achieved by CortiLife is already quite promising.
> >
> > | Setting          | CT ACC        | CT AUC        | MC ACC        | MC AUC        | SD ACC        | SD AUC        |
> > |------------------|---------------|---------------|---------------|---------------|---------------|---------------|
> > | BatchNorm+Linear | 0.645±0.049   | 0.664±0.024   | 0.597±0.021   | 0.595±0.057   | 0.591±0.039   | 0.603±0.037   |
> > | LayerNorm+Linear | 0.626±0.018   | 0.629±0.043   | 0.611±0.007   | 0.599±0.045   | 0.622±0.033   | 0.618±0.034   |
> > | 0.75 (Ours)      | **0.697±0.015** | **0.730±0.007** | **0.632±0.005** | **0.618±0.018** | **0.651±0.010** | **0.657±0.017** |
> >
> > [1]Zhang S, Jiang Z, Shen X, et al. Graph Disentanglement Learning for fMRI Analysis: Decoupling Disease, Covariates, and Individual Variability[C]//International Conference on Medical Image Computing and Computer-Assisted Intervention. Cham: Springer Nature Switzerland, 2025: 352-361.
> >
> > [2]Hu X, Wang W, Xiao L. Learning 3D Medical Image Models from Brain Functional Connectivity Network Supervision for Mental Disorder Diagnosis[C]//International Conference on Medical Image Computing and Computer-Assisted Intervention. Cham: Springer Nature Switzerland, 2025: 336-346.
> >
> > ## Q7. CARZero Implementation Verification
> >
> > First, we confirm that the implementation of CARZero is correct. Its code is directly taken from the official repository of the original paper, and its performance on disease classification tasks matches that reported for other CLIP based methods.
> >
> > Second, we explain why CARZero performs poorly on brain parcellation in our setting. CLIP and ACLIP, including our method, align a global visual [CLS] token with a global textual [CLS] token, so this alignment does not interfere with the learning of local patch representations needed for cortical parcellation. By contrast, the radiology reports used by CARZero usually contain explicit anatomical region descriptions, and its cross modal alignment is designed to match local image patches with text that carries region specific semantics. In our task, however, the text prompts provide only subject level attributes such as age and sex and do not contain any spatially resolved information that can be assigned to specific cortical patches. This global semantic signal is fundamentally mismatched with the local alignment mechanism of CARZero, which leads to a severe drop in performance. This behavior reflects an intrinsic incompatibility between the original architecture and the nature of our task, rather than an implementation error. Following the reviewer’s suggestion, we have removed CARZero from the comparative baselines for the brain parcellation task in the revised manuscript in order to avoid potentially misleading side by side comparisons.

---

### Official Review · Reviewer_o3ur · 2025-10-29

**Soundness:** 4
**Presentation:** 3
**Contribution:** 3
**Rating:** 6
**Confidence:** 3

**Summary:**

This paper presents a framework for extracting information from cortical mesh data. Among the information, there is the reconstruction of the cortical surface and metadata about the subject. The experiments are done using several classical datasets in neuroscience.

**Strengths:**

The framework extract much information from surface cortical mesh with very important metadata. The text is clear and the model well presented. The experiments show interesting results. While the model is not always the best, it is the most reliable. The potential impact of such in the neuroscience community is great since the cortical surface important information about the health of the subject.

**Weaknesses:**

I see several weaknesses in such paper.

- There exists strong fairness issues with cortical data, especially on the gender. Do such biases affect the model?
- I am not sure of the impact of such in the ICLR community. While such framework can have a large audience in the neuroscience community, it is not clear that the classical machine learning community will be intereted. Especially there exists now several frameworks of this kind in medical sciences (see [1]).
- Extracting the age and gender using a LLM seems a little overkill and misleading. Why not using a regression model?
- Similar frameworks have been proposed for other kind of medical imaging (for example, see [1, 2, 3]). What is the position of this model compared to these in terms of capabilities?

### References

- [1] Khan, W., Leem, S., See, K. B., Wong, J. K., Zhang, S., & Fang, R. (2025). A comprehensive survey of foundation models in medicine. IEEE Reviews in Biomedical Engineering.
- [2] Lu, M. Y., Chen, B., Williamson, D. F., Chen, R. J., Liang, I., Ding, T., ... & Mahmood, F. (2024). A visual-language foundation model for computational pathology. Nature medicine, 30(3), 863-874.
- [3] Zhang, K., Zhou, R., Adhikarla, E., Yan, Z., Liu, Y., Yu, J., ... & Sun, L. (2024). A generalist vision–language foundation model for diverse biomedical tasks. Nature Medicine, 30(11), 3129-3141.

**Questions:**

See the weakness section for the questions.

---

> ### Author Response · Authors · 2025-11-22
> **Thank the reviewer for the constructive suggestions. We have provided detailed responses to all comments.**
>
> # Response to Reviewer 2
> We appreciate your positive comments on our work, including “well presented”, “interesting results” and “great potential impact” etc. Detailed responses to all comments are provided below.
> ---
> ## (1)W1 There exists strong fairness issues with cortical data, especially on the gender. Do such biases affect the model?
> We fully agree with that gender is a key factor contributing to cortical structural heterogeneity. Numerous studies in neuroscience have demonstrated pronounced differences between male and female brains in cortical morphology. Therefore, explicitly modeling gender information is essential for learning accurate cortical representations. Motivated by this, we incorporate gender information into the Metadata Language Prompting during model design, enabling the model to explicitly learn gender-related anatomical variation rather than treating it as noise. As shown in the visualization in Figure 4(b), the learned representations naturally separate into two clusters corresponding to male and female subjects, which reflects true biological differences. Meanwhile, as shown in Figure 4(c), the disease group and the healthy control group remain clearly separated into two distinct clusters even within the same gender cluster. This observation indicates that the model indeed captures genuine disease-related structural differences rather than being driven by gender-related biases.
>
> Furthermore, to evaluate the impact of removing gender information, we conducted an additional ablation study in which gender was excluded from the Metadata Language Prompting. We retrained CortiLife and assessed its performance on the ADHD classification task. As shown in the table below, removing gender leads to a noticeable drop in accuracy. This is expected, as natural gender-related variations become entangled with disease-related differences, thereby making the classification task more challenging. We have incorporated these additional results into Table 8 in the appendix of the final version.
>
> | Setting  | CT ACC        | CT AUC        | MC ACC        | MC AUC        | SD ACC        | SD AUC        |
> |:----------|:---------------:|:---------------:|:---------------:|:---------------:|:---------------:|:---------------:|
> | w/o Sex  | 0.687±0.005   | 0.703±0.006   | 0.621±0.019   | **0.662±0.021**   | 0.629±0.036   | 0.635±0.047   |
> | w/ Sex   | **0.697±0.015** | **0.730±0.007** | **0.632±0.005** | 0.618±0.018   | **0.651±0.010** | **0.657±0.017** |

---

> > ### Author Response · Authors · 2025-11-22
> >
> > ## (2) W2&W4 I am not sure of the impact of such in the ICLR community. While such framework can have a large audience in the neuroscience community, it is not clear that the classical machine learning community will be intereted. Especially there exists now several frameworks of this kind in medical sciences (see [1])：
> >
> > Regarding the reviewer’s concern about “community interest,” we believe that our work is highly relevant to the ICLR community and can make meaningful contributions to research at the intersection of machine learning and neuroscience. In recent years, several neuroscience-related foundation models have been published at ICLR, such as LaBraM (ICLR 2024, 204 citations) [1] and NeuroLM (ICLR 2025, 49 citations) [2], and they have received significant attention, indicating the community’s growing interest in brain-inspired and neural-signal representation learning. In addition, the core contributions of our work fundamentally lie within machine learning. We address a broadly applicable and foundational ML problem: representation learning on non-Euclidean manifolds. The cortical surface is a representative and challenging instance of this setting. Its highly folded geometry and complex curvature naturally give rise to several classical ML questions, including geometric encoding on curved manifolds, structure-aware feature learning, and understanding how heterogeneous factors such as age and gender influence the learned representation space.
> >
> > Regarding the reviewer’s concern that “several frameworks of this kind already exist in the medical domain (see [1])”, we would like to clarify that existing medical vision-language models differ fundamentally from the problem setting addressed in our work, and therefore do not diminish the novelty or necessity of our contribution. First, unlike natural images, medical images differ drastically across modalities. For example, the reviewer-mentioned CONCH [3] is specifically designed for histopathology, operating on ultra-high-resolution whole-slide images with fine-grained cellular textures. BiomedGPT [4] focuses on 2D Euclidean medical images such as CT slices, X-rays, dermoscopy, and fundus photographs. These data types lie on regular pixel grids and do not involve complex geometric structures(such as cortical surface). Second, although some survey papers list MRI-related models, these models are almost exclusively based on 2D slice-level Euclidean representations rather than on cortical surface representation learning. In contrast, the cerebral cortex is a highly folded non-Euclidean 2D manifold, and learning geometry-aware multimodal representations directly on this surface introduces modeling challenges that do not arise in standard 2D/3D medical imaging. To the best of our knowledge, no existing foundation model has been developed specifically for cortical surface data. Finally, cortical representation learning is a core problem in computational neuroscience. Cortical features provide key insights into brain topology and play an essential role in disease detection, developmental analysis, and mechanistic understanding. Addressing this unique domain is therefore both scientifically important and methodologically distinct from existing medical foundation models.
> >
> > [1] Jiang W B, Zhao L M, Lu B L. Large brain model for learning generic representations with tremendous EEG data in BCI[J]. arXiv preprint arXiv:2405.18765, 2024.
> >
> > [2] Jiang W B, Wang Y, Lu B L, et al. NeuroLM: A universal multi-task foundation model for bridging the gap between language and EEG signals[J]. arXiv preprint arXiv:2409.00101, 2024.
> >
> > [3] Lu M Y, Chen B, Williamson D F K, et al. A visual-language foundation model for computational pathology[J]. Nature medicine, 2024, 30(3): 863-874.
> >
> > [4] Zhang K, Zhou R, Adhikarla E, et al. A generalist vision–language foundation model for diverse biomedical tasks[J]. Nature Medicine, 2024, 30(11): 3129-3141.

---

> > > ### Author Response · Authors · 2025-11-22
> > >
> > > ## (3) W3 Extracting the age and gender using a LLM seems a little overkill and misleading. Why not using a regression model?
> > > Our goal is not to “predict age or gender” from cortical data, but rather to leverage metadata as stable, high-level semantic anchors to enhance the quality of cortical representations across age ranges and modalities. Under this objective, a simple regression model is inherently insufficient.
> > >
> > > First, cortical structural signals (e.g., CT, MC, SD) encode complex developmental and disease-related semantics that extend far beyond low-dimensional numeric attributes(like age and sex prediction). PubMedBERT, pretrained on large-scale biomedical corpora, provides a rich, semantically coherent embedding space in which diverse metadata can be interpreted and integrated. Unlike a regression model that outputs only a scalar value, an LLM encodes multi-dimensional semantic relations and thereby guides the visual encoder toward representations that better capture high-level cortical morphology.
> > >
> > > Second, the metadata we incorporate are not limited to age and gender. These two variables function merely as proxy information. Our textual input additionally includes health status and cortical feature type, among other factors. Such heterogeneous metadata cannot be adequately expressed or jointly modeled using a simple regression framework, whereas an LLM can naturally embed these elements into a unified semantic representation.
> > >
> > > In summary, the use of LLM is not intended to “extract” age and gender, but to construct a semantically aligned biomedical embedding space that enhances the robustness and generalization ability of our cortical representation learning framework, which that cannot be achieved with a simple regression model.

---

### Official Review · Reviewer_gb7Z · 2025-10-30

**Soundness:** 3
**Presentation:** 3
**Contribution:** 3
**Rating:** 6
**Confidence:** 3

**Summary:**

The authors present a method called CortiLife, a unified vision-language framework for lifespan-aware cortical representation learning. They design a surface tokenizer to capture local topology, global interactions, and patch-wise distributional patterns. During training, CortiLife uses masked self-distillation to avoid substantial information redundancy and metadata language prompting to embed extra information such as age, sex and so on.

**Strengths:**

- The method can handle data from any lifespan.
- The authors build a surface tokenizer and training framework tailored to cortical data.
- The method outperforms existing methods, indicating potential practical application value.

**Weaknesses:**

- In Table 2, CARZero’s DICE performance appears surprisingly poor—can authors provide an explanation?
- The “zero-shot” experiments seem to require additional training; doesn’t this resemble “pretraining + linear probe” rather than conventional zero-shot generalization?
- The authors could consider to add more experiments on ablation study—for example, varying the masking ratio and whether to include $L_{CLIP}$—to more comprehensively show each module’s marginal contribution.

**Questions:**

Three encoding types in Figure 2 can be drawn more specifically to facilitate readers' understanding

---

> ### Author Response · Authors · 2025-11-22
> **Thank the reviewer for the constructive suggestions. We have provided detailed responses to all comments.**
>
> # Response to Reviewer 1
> We sincerely appreciate the reviewers’ constructive and positive feedback. Below, we provide detailed responses to all comments.
> ---
> ## (1)W 1: In Table 2, CARZero’s DICE performance appears surprisingly poor—can authors provide an explanation?
> The notably low DICE scores of CARZero in Table 2 primarily stem from a semantic granularity mismatch between its alignment mechanism and the cortical parcellation task. CARZero was originally developed for chest X-ray disease classification, where radiology reports contain region-specific textual descriptions. Accordingly, its alignment strategy focuses on matching local image patches with localized textual semantics. In our setting, however, the text prompts encode only global subject-level attributes (e.g., age, sex) and do not provide any spatially resolved information. These global semantics cannot correspond to specific cortical patches. During cross-attention alignment, the global textual signal dominates, causing patch-level visual representations to collapse toward a uniform global embedding and eliminating the local structural variability required for parcellation. Since cortical parcellation relies on fine-grained, region-specific morphological distinctions, this collapse severely impairs the model’s ability to differentiate cortical areas, resulting in CARZero’s extremely low DICE performance. In contrast, CLIP-based methods (CLIP, ACLIP, DetailCLIP) employ global vision-text alignment and thus avoid the local-global semantic mismatch, achieving substantially better results despite still being inferior to our approach.
>
> ## (2) W2 The “zero-shot” experiments seem to require additional training; doesn’t this resemble “pretraining + linear probe” rather than conventional zero-shot generalization? ：
> We agree that, under the terminology commonly used in the CLIP and foundation model literature, our current evaluation protocol does not correspond to a strict zero-shot setting. Although all parameters of the cortical vision encoder remain fully frozen during downstream evaluation, a shallow MLP head must still be trained for each specific downstream task. For example, a regression head is required for age prediction. Therefore, this protocol aligns more closely with a “frozen-encoder + linear evaluation” setting rather than a traditional zero-shot setup.
>
> To avoid ambiguity, we have updated the terminology in the final version of the paper and renamed the two evaluation paradigms as “**Encoder Frozen on Downstream Tasks**” and “**Encoder Fine-tuning on Downstream Tasks**”. In the Encoder Frozen on Downstream Tasks setting, the cortical encoder is completely frozen and only the task-specific MLP head is trained. This setting is intended to assess the cross-dataset transferability and generalization ability of the learned cortical representations. In the Encoder Fine-tuning on Downstream Tasks setting, both the cortical encoder and the task-specific MLP head are jointly fine-tuned on the downstream dataset, which evaluates how well the learned representations adapt is permitted.

---

> ### Author Response · Authors · 2025-11-22
> **Thank the reviewer for the constructive suggestions. We have provided detailed responses to all comments.**
>
> ## (3) W3 The authors could consider to add more experiments on ablation study-for example, varying the masking ratio and whether to include $L_{CLIP}$-to more comprehensively show each module’s marginal contribution.
> Thanks. We have added two additional ablation studies: 1) varying the masking ratio across 0.5, 0.6, 0.75, 0.8, and 0.9 to systematically analyze its impact on the learned representations, and 2) comparing the model performance with and without the $L_{CLIP} loss to verify its effectiveness. All experiments were performed on the ADHD classification task, where we retrained CortiLife under the “Encoder Fine-tuning on Downstream Tasks” setting. All newly added results have been included in the appendix of the revised paper.
>
> Table below presents the results for different masking ratios. We observe that a masking ratio of 0.75 consistently yields the best overall performance across nearly all tasks. All newly added results have been included in the appendix of the revised paper.
>
> | Mask ratio |        CT ACC        |       CT AUC        |        MC ACC        |       MC AUC        |        SD ACC        |       SD AUC        |
> |:------------|:-----------------------:|:----------------------:|:-----------------------:|----------------------:|:-----------------------:|:----------------------:|
> | 0.5        | 0.681±0.010           | 0.697±0.031          | 0.621±0.018           | 0.632±0.034          | 0.631±0.015           | 0.648±0.029          |
> | 0.6        | 0.683±0.028           | 0.706±0.011          | 0.630±0.009           | 0.619±0.025          | 0.617±0.021           | 0.607±0.014          |
> | 0.8        | 0.671±0.034           | 0.708±0.053          | 0.628±0.008           | 0.658±0.010          | 0.628±0.030           | 0.604±0.046          |
> | 0.9        | 0.667±0.027           | 0.713±0.011          | 0.618±0.037           | **0.687±0.012**      | 0.598±0.018           | 0.609±0.036          |
> | 0.75 (Ours)| **0.697±0.015**       | **0.730±0.007**      | **0.632±0.005**       | 0.618±0.018          | **0.651±0.010**       | **0.657±0.017**      |
>
> Table below reports the ablation results for $L_{CLIP}$, showing that incorporating this loss further improves the quality of the learned representations.
>
> | Setting        | CT ACC        | CT AUC        | MC ACC        | MC AUC        | SD ACC        | SD AUC        |
> |:----------------|:---------------:|:---------------:|:---------------:|:---------------:|:---------------:|:---------------:|
> | w/o $L_{CLIP}$     | 0.630±0.005   | 0.603±0.008   | 0.609±0.013   | 0.585±0.011   | 0.605±0.013   | 0.566±0.028   |
> | w/ $L_{CLIP}$      | **0.697±0.015** | **0.730±0.007** | **0.632±0.005** | **0.618±0.018**   | **0.651±0.010** | **0.657±0.017** |
>
> ## (4) Q1 Three encoding types in Figure 2 can be drawn more specifically to facilitate readers' understanding.
>
> We have revised Figure 2 to make the three encoding components more explicit and easier to understand. Specifically: 1) In the Local Topology Encoding module (Fig. 2(b1)), we provide a more direct illustration of the internal structure within each patch and clarify how this structure is processed using spherical convolution to obtain the corresponding local topology encoding. 2) In the Global Interaction Encoding module (Fig. 2(b2)), we elaborate on how the original cortical surface is partitioned into patches and how patch interaction representations are captured using self-attention followed by feedforward block. 3) In the Patch-wise Distribution Encoding module (Fig. 2(b3)), we refined the visualization to highlight the output of the scale-adaptive encoder, emphasizing how heterogeneous age-related distributions are projected into shared multi-scale statistical spaces, thereby mitigating distribution shifts introduced by age differences.

---

### Meta-Review · Area_Chair_RZtu · 2026-01-06

**Summary:**

The paper presents "CortiLife," a unified vision-language foundation model designed for cortical representation learning across the human lifespan (from 26 weeks to 95 years). The framework introduces a multi-level surface tokenizer for cortical meshes and employs a masked self-distillation strategy combined with metadata language prompting (age, sex, etc.) via a teacher-student architecture. Reviewers acknowledged the impressive data collection effort (over 13,000 subjects), the technical soundness of the surface patchification approach, and the competitive empirical performance across multiple downstream tasks, noting its potential impact on the neuroscience community.

**Reviewer Concerns:**

The initial review process raised critical concerns regarding "circular reasoning" (specifically whether age/gender clusters were merely reflections of text inputs) and whether performance gains were driven by data scale rather than architectural innovation. In the rebuttal, the authors effectively addressed these points by:

Providing new experiments showing that the model maintains discriminative feature representations even when metadata prompts are withheld during inference, thus debunking the circular reasoning concern.

Presenting detailed ablation studies that isolate the contributions of the scale-adaptive encoder and the masked self-distillation mechanism, proving their value over simple data aggregation.

Clarifying baseline pretraining protocols to ensure a fair "apples-to-apples" comparison and justifying the 75% masking ratio through spatial autocorrelation analysis specific to cortical anatomy.

**Reviewer Scores:**

The paper received scores of 6, 6, and 4. While Reviewer Hj8o (4) initially presented a high-confidence critique of the methodology, the Area Chair (AC) finds that the author’s comprehensive response has successfully resolved the most significant technical doubts. The other two reviewers (6, 6) maintained their positive stance, emphasizing the framework's reliability and its breakthrough role as a lifespan-aware foundation model. Following the discussion and the provided evidence, the AC is satisfied that the paper’s empirical robustness and original tokenizer design justify its acceptance.

---

### Decision · Program_Chairs · 2026-01-26

Accept (Poster)